# STGAD: Self-temporal generative adversarial framework with transformer attention for unsupervised multivariate time-series anomaly detection and localization

Xiao Liao[1], Wei Deng[1], Hongyue Ma[1], Yihan Mu[2]*

**1** State Grid Information and Telecommunication Group Co., Ltd., Beijing, China, **2** School of Automation Science and Engineering, Xi'an Jiaotong University, Xi'an, China

* 1710491817@qq.com

## Abstract

Unsupervised anomaly detection in multivariate time series is important for maintaining the reliability of complex cyber-physical systems. However, existing methods often face practical challenges in adversarial stability, temporal dependency modeling, and anomaly-score calibration across datasets. We present STGAD, a dual-score generative-adversarial framework for anomaly detection and localization in multivariate time series. STGAD employs a WGAN-GP critic with a Transformer encoder to perform self-temporal modeling of within-window dependencies and cross-variable interactions, and uses a stochastic generator trained under adversarial supervision with sample-level proximity regularization to model normal temporal patterns. During inference, multiple generated candidates are sampled for each input window, and the minimum residual is used as a sample-matching anomaly cue. This residual-based score is fused with the critic-based score after normalization, and final anomaly decisions are produced by distribution-adaptive thresholding. Experiments on five benchmark datasets spanning server monitoring, aerospace telemetry, industrial control, and ECG signals (SMD, SMAP, MSL, SWaT, and MIT-BIH) show that STGAD achieves strong and consistent performance against representative baselines. Ablation and robustness analyses further demonstrate the effectiveness of critic-side temporal modeling, stable adversarial learning, and dual-score fusion in the proposed framework.

## 1. Introduction

Modern IT operations and industrial IoT ecosystems continuously generate massive, high-dimensional, and strongly correlated streams of sensor readings and logs at sub-second cadence to monitor system health, service performance, and security risk in large-scale infrastructures [1,2]. Traditionally, engineers and data-mining

**Data availability statement:** Data were obtained from third-party public repositories and controlled-access sources. The SMD (Server Machine Dataset) is available at the OmniAnomaly repository: https://github.com/NetManAIOps/OmniAnomaly. The SMAP and MSL (NASA Telemetry Datasets) are available at the NASA telemetry repository: https://s3-us-west-2.amazonaws.com/telemanom/data.zip. The SWaT (Secure Water Treatment Dataset) is available upon request from iTrust, Singapore University of Technology and Design, at: https://itrust.sutd.edu.sg/test-beds/secure-water-treatment-swat/. The MIT-BIH Arrhythmia Database is available from PhysioNet at: https://physionet.org/content/mitdb/1.0.0/. The code implementation of STGAD is publicly available at: https://github.com/shy3119/STGAD.git.

**Funding:** This work was supported in part by the State Grid Information and Telecommunication Group Co., Ltd., which coordinates scientific and technological projects (SGIT0000XTJS2401078). There was no additional external funding received for this study. The funder provided support in the form of salaries for authors X.L., W.D. and H.M., but did not have any additional role in the study design, data collection and analysis, decision to publish, or preparation of the manuscript.

**Competing interests:** The authors have declared that no competing interests exist.

**Abbreviation:** STGAD: Self-Temporal Generative Adversarial Detector, MTS-AD: multivariate time-series anomaly detection, GAN: Generative Adversarial Network, WGAN-GP: Wasserstein GAN with Gradient Penalty, MLP: Multi-Layer Perceptron, RNN: Recurrent Neural Network, CNN: Convolutional Neural Network, VAE: Variational Autoencoder, ELBO: Evidence Lower Bound, GNN: Graph Neural Network, SMD: Server Machine Dataset, SMAP: Soil Moisture Active Passive (NASA dataset), MSL: Mars Science Laboratory (NASA telemetry dataset), SWaT: Secure Water Treatment (testbed dataset), MIT-BIH: MIT-BIH Arrhythmia dataset, AUC: Area Under the Receiver Operating Characteristic Curve, PR-AUC: Area Under the Precision–Recall Curve: P: Precision, R: Recall, F1: F1-score (harmonic mean of Precision and Recall), POT: Peak-Over-Threshold, CPU: Central Processing Unit, ECG: Electrocardiogram.

practitioners have relied on rule bases, statistical control charts, and expert thresholds to flag deviations from nominal behavior and issue fault reports—artifacts that underpin reactive fault tolerance and robust database/system design workflows [3,4]. As cloud-native platforms, edge–cloud pipelines, and automated production lines scale, the dimensionality, sampling rates, and throughput of telemetry increase sharply, while anomaly manifestations diversify (point, contextual, collective; abrupt and gradual), making single rules or static statistics insufficient to meet simultaneous requirements of low miss rate and low false alarm in online monitoring [1,5]. The maturation of big-data analytics and deep learning has therefore brought renewed attention—across data mining and AIOps communities—to the problem of automatically discovering departures from normal mechanisms under weak or absent labels in high-dimensional time series. In Industry-4.0 manufacturing and process control, scenarios centered on service reliability and autonomous fault management are particularly salient: unified modeling of sensor streams and event logs enables early detection of incipient anomalies, triggering failover and self-healing to reduce downtime and maintenance cost [5–7]. This problem is commonly formulated as multivariate time-series anomaly detection (MTS-AD): identifying observations or segments that deviate from expected temporal evolution in high-dimensional, non-stationary sequences that may exhibit long-range dependencies [8,9]. Leveraging scalable computation and streaming frameworks, data-driven sectors—including distributed computing, IoT/vehicular systems, robotics and process industries, as well as urban infrastructure for energy and transportation—are increasingly adopting machine-learning-based unsupervised/self-supervised methods to improve alert timeliness and accuracy: one line learns normal behavior and measures deviation via reconstruction/prediction residuals; another performs explicit or implicit density modeling to estimate sample-level likelihood; still others employ adversarial or representation learning to better capture complex inter-variable structure and weak-magnitude anomalies [10,11]. This shift from expert-driven heuristics to data-driven modeling provides a practical pathway toward predictive maintenance and strengthened security in large-scale operational systems [12–14].

As modern telemetry scales across devices and services, multivariate time-series anomaly detection is becoming increasingly difficult due to higher dimensionality, richer modalities, and greater volatility. Contemporary IoT/IT platforms continually add sensors and microservices, inflating correlation structure and sampling heterogeneity, which in turn raises the data requirements for reliable inference and robust calibration. Meanwhile, privacy constraints and the rise of federated/geo-distributed training make cross-site synchronization costly, yielding fragmented, non-IID local datasets that hinder generalizable representation learning. Because these time series originate from engineered systems interacting with humans and environments, observations exhibit both stochastic variability and structured temporal dynamics; models must disentangle random fluctuations from genuine deviations under non-stationarity, concept drift, long-range dependencies, and time-varying inter-variable couplings [11,15]. Moreover, label scarcity and anomaly diversity limit the direct use of supervised paradigms that succeed elsewhere in data mining, while contamination of the

"normal" set by a small number of anomalies weakens unsupervised objectives and thresholding [16,17]. Finally, practical value hinges not only on detecting anomalies but also on root-cause localization—identifying which variables, components, or subsystems drive failures—turning the task into joint detection and attribution under distribution shift and sparse feedback, which further complicates methodology and evaluation [18,19].

Existing approaches to multivariate time-series anomaly detection (MTS-AD) fall into several families. Statistical/signal-processing methods—control charts, change-point tests, ARIMA/Kalman filtering, and spectral/wavelet analysis—offer lightweight, interpretable pipelines grounded in explicit stochastic assumptions, yet struggle with high-dimensional nonlinearity and time-varying couplings [20–23]. Classical machine-learning techniques (e.g., OC-SVM, Isolation Forest, LOF and streaming variants) quantify outliers via density/distance or margin-based criteria, working well in moderate dimensions or weakly temporal settings but lacking expressivity for long-range dependencies and inter-variable structure [24–27]. Deep learning advances have proceeded along two main lines: reconstruction/prediction-driven models (auto-encoders/VAEs, TCN/LSTM, Seq2Seq forecasting) that learn nominal dynamics and score deviations via residuals; and probabilistic/ generative models (normalizing flows, diffusion, GANs) that estimate sample likelihoods or discriminator confidences from explicit or implicit distributions [11,13,14]. With the rise of attention mechanisms, Transformer-based architectures strengthen long-horizon temporal modeling, while graph neural/attention models encode cross-variable relations using prior or adaptive graphs; hybrid designs combine temporal and graph structure for greater expressivity, and recent engineering AI studies have also highlighted the value of structured relational reasoning for learning from complex heterogeneous data [28–31]. On the decision side, researchers explore multi-score fusion, distribution-adaptive thresholds (e.g., EVT/POT), segment-level post-processing, and calibration to reduce false alarms and improve cross-domain comparability [32,33]. Practicality is further supported by work on online/incremental learning, knowledge distillation/pruning, federated learning, and explainable/causal attribution [34]. Despite these advances, notable gaps remain: robust detection of weak-magnitude and long-period anomalies; reliable score calibration under non-stationarity and distributional shifts; and cross-scenario generalization. Reconstruction/prediction objectives can "absorb" mild anomalies and compress decision margins; generative/adversarial objectives face training stability and score-scale consistency issues; and attention/graph models often rely on windowing or fixed priors that become brittle when transferred.

Building on sequence modeling and adversarial learning, we propose STGAD, a dual-score generative-adversarial framework for unsupervised multivariate time-series anomaly detection. STGAD combines a WGAN-GP critic with a Transformer encoder for within-window temporal modeling and a stochastic generator trained with adversarial and sample-level proximity constraints. During inference, the generator produces multiple candidate windows, and the minimum residual to the observed window is used as a sample-matching anomaly cue. This residual-based signal is fused with the critic-based score after validation-based normalization, and anomaly decisions are obtained through distribution-adaptive thresholding. Extensive experiments on five benchmark datasets show that this design provides strong and consistent performance across heterogeneous scenarios. Overall, STGAD offers an effective combination of temporal expressivity, adversarial stability, score calibration, and practical applicability for anomaly detection and localization in multivariate time series. The remainder of this paper is organized as follows. Section 2 reviews representative studies on multivariate time-series anomaly detection. Section 3 presents the proposed STGAD framework, including the model architecture and the training/inference procedure. Section 4 describes the datasets, preprocessing steps, model configuration, and evaluation protocol. Section 5 reports the comparative results together with ablation, robustness, sensitivity, and interpretability analyses. Finally, Section 6 concludes the paper and outlines directions for future work.

## 2. Related work

Traditional anomaly detection methods typically model the distribution of time-series data in an unsupervised manner. One major line is clustering (e.g., k-means and regression-style scoring), which identifies "normal" prototypes and flags deviations via distances or within-cluster residuals [35]; another line comprises distance-based schemes (fixed-radius counts,

k-NN distance ranks) that trigger alarms when local neighborhoods thin out [36]. In parallel, density models (e.g., kernel density estimation or mixture models) label low-likelihood samples using (log-)probability thresholds [37], while isolation methods (Isolation/half-space trees) recursively partition the feature space so that points that are easier to isolate receive higher anomaly scores [38]. To better exploit temporal structure, signal-transform techniques—most notably wavelet and Hilbert transforms—map sequences into time–frequency or instantaneous amplitude/phase domains, from which robust multiscale features are extracted and fed to the above scorers [39]. A complementary autoregressive forecasting–residual paradigm (AR/ARMA/ARIMA/SARIMA, and ARIMAX when exogenous covariates are available) models short-range autocorrelation and seasonality and then uses one- or multi-step prediction errors as anomaly scores [40]. However, these classical routes generally rely on near-stationarity and weak nonlinearity: in high-order, strongly coupled, and non-stationary multivariate telemetry, distance/density methods become sensitive to metric choice and bandwidths with poor threshold transferability; clustering and density models require frequent re-training under concept drift; transform-domain methods depend on band/scale selection; and linear AR models struggle with long-range dependencies, time-varying inter-variable couplings, and weak-magnitude/contextual/collective anomalies. As a result, classical techniques often need to be combined with more expressive temporal representations and calibration strategies in modern operational settings.

For high-dimensional, tightly coupled telemetry, modern MTS-AD models replace hand-crafted scores with learned reconstructions, likelihoods, and relational reasoning. DAGMM [41] couples a deep autoencoder with a latent Gaussian-mixture estimator: the decoder minimizes reconstruction error while the GMM yields an energy score unifying residual and likelihood; however, near-Gaussian latent assumptions can underfit multi-modal normality and hinder cross-dataset calibration. OmniAnomaly [11] employs a stochastic recurrent VAE with temporal priors to capture non-stationary dynamics and uncertainty, detecting via reconstruction probability and KL terms; training/scoring can be sensitive to sequence length and hyperparameters, and mild anomalies may be "absorbed" by the decoder. MAD-GAN [14] uses recurrent generators/discriminators so that discriminator confidence and reconstruction discrepancy jointly expose outliers, yet adversarial optimization can be unstable (e.g., mode collapse) and is often confined to short windows. MSCRED [42] constructs multi-scale signature matrices (variable–variable correlation snapshots) and applies convolutional encoder–decoders to reveal deviations across time and inter-series structure; performance depends on window/scale choices and may miss weak, long-period drifts. USAD [33] adopts a dual-autoencoder scheme trained with complementary reconstruction objectives (and a minimax-style loss on first/second-stage reconstructions) to produce more robust residual scores under contamination; while efficient and architecture-simple, it remains window-based, can still compress weak anomalies into the reconstruction channel, and requires careful threshold calibration across datasets. TranAD [28] leverages a Transformer encoder–decoder with self-conditioning and adversarial training to stabilize long input windows and amplify small residuals; nevertheless, attention is typically bounded to a fixed window and score calibration still needs extra care. GDN [30] learns an adaptive variable graph and applies graph attention to quantify node-level deviations from learned relations—well suited to time-varying couplings—yet learned graphs are noise-sensitive, erroneous edges inflate false alarms, and graph construction/inference costs grow with dimensionality. Overall, these models advance expressivity across reconstruction/likelihood, adversarial discrimination, and temporal-graph structure, while leaving open issues in weak-magnitude/long-period detectability, robust calibration, and brittleness to windowing or priors.

Table 1 summarizes the strengths and limitations of mainstream methods: DAGMM performs representation compression via an autoencoder coupled with a Gaussian mixture model but exhibits limited temporal modeling; OmniAnomaly employs a variational recurrent architecture to capture temporal uncertainty, yet is sensitive to noise; MAD-GAN adopts a generative adversarial framework with recurrent units to model nonlinear dynamics, but suffers from training instability; MSCRED extracts multi-scale features using convolutional and recurrent networks at a higher computational cost; TranAD introduces local self-attention with a local Transformer but provides restricted global context; GDN encodes inter-variable relations with graph neural networks, although it often relies on prior graph structure.

**Table 1. Comparison of representative methods in terms of key ideas, advantages, and limitations.**

| Method | Key Ideas | Advantages | Limitations |
|---|---|---|---|
| DAGMM | Autoencoder + GMM | Dimensionality reduction | Weak temporal modeling |
| OmniAnomaly | Variational RNN | Temporal uncertainty | Noise sensitivity |
| MAD-GAN | GAN + LSTM | Nonlinear dynamics | Training instability |
| MSCRED | Multi-scale CNN + RNN | Multi-scale features | High computation cost |
| TranAD | Local transformer | Local attention | Limited global context |
| GDN | Graph neural networks | Graph relations | Prior graph required |
| STGAD (proposed) | Dual-score GAN | Global temporal modeling | Sampling overhead |

STGAD addresses these gaps through a dual-score generative-adversarial design for multivariate time-series anomaly detection. The framework combines a WGAN-GP critic with a Transformer encoder to capture within-window temporal dependencies and cross-variable interactions, together with a stochastic generator that provides complementary residual evidence through sample matching at inference time. By fusing the critic-based score and the residual-based score under validation-based normalization and distribution-adaptive thresholding, STGAD enhances sensitivity to weak, gradual, and contextual anomalies while improving score stability across datasets. Overall, the framework provides an effective balance of temporal modeling, adversarial robustness, and practical anomaly scoring without relying on external graph priors.

## 3. Methodology

This section presents the detailed methodology of the proposed STGAD framework. We begin by formulating the multivariate time-series anomaly detection problem and describing how the data is segmented for training. We then introduce the overall model architecture, including the design of the Generator and Discriminator components. Finally, we outline the training and inference procedures, highlighting the use of WGAN-GP and the dual scoring mechanism for anomaly detection.

### 3.1. Problem formulation

Let $X = \{x_1, x_2, \ldots, x_T\}$ be a multivariate time series, where each time step $x_t \in \mathbb{R}^d$ denotes a d-dimensional observation vector collected from a system with multiple sensors. The goal of anomaly detection is to determine whether a given subsequence exhibits behavior that deviates significantly from the normal patterns observed in the historical data.

To capture local temporal dependencies and generate structured inputs for training, we segment the time series into overlapping fixed-length windows of size $w$. Each window is defined as:

$$W_t = \{x_{t-w+1}, x_{t-w+2}, \ldots, x_t\}, \quad x_i \in \mathbb{R}^d \tag{1}$$

The resulting training set is a sequence of windows $W = \{W_1, W_2, \ldots, W_{T'}\}$, where $T' = T - w + 1$.

We assume that the training dataset consists of primarily normal behavior, with negligible or no anomalous data. The model is trained unsupervised, learning the distribution of normal patterns. During inference, given a test window $\hat{W}_t$, the model computes an anomaly score $A\left(\hat{W}_t\right) \in \mathbb{R}_{\geq 0}$, and compares it to a predefined threshold $\lambda$ to determine whether an anomaly has occurred:

$$y_t = \begin{cases} 1, & \text{if } A\left(\hat{W}_t\right) \geq \lambda \\ 0, & \text{otherwise} \end{cases} \tag{2}$$

where $y_t$ = 1 denotes an anomaly and $y_t$ = 0 indicates normal behavior.

Modeling time series at the window granularity enables us to encode temporal evolution together with cross-variable dependencies—capabilities that are crucial for uncovering subtle and complex anomalies in multivariate settings.

## 3.2. Model architecture

STGAD adopts a dual-score generative–adversarial framework for multivariate time-series anomaly detection. The framework consists of a stochastic Generator and a WGAN-GP-based Discriminator. During training, the Generator synthesizes plausible temporal windows under adversarial supervision together with a sample-level proximity constraint, while the Discriminator learns to distinguish observed windows from generated ones. During inference, anomaly evidence is derived from two complementary signals: a critic-based score from the Discriminator and a sample-matching residual obtained from multiple generated candidates. Fig 1 illustrates the overall architecture and information flow of the framework.

Generator Architecture. The Generator in STGAD is a stochastic temporal generator that maps latent noise sequences to plausible multivariate windows. As shown in Fig 2, a sequence of latent vectors is first projected into a shared hidden space, then processed by an LSTM-based temporal encoder to capture sequential dependencies, and finally decoded step by step by a two-layer MLP to produce the multivariate output sequence. In this way, the Generator learns structured temporal dynamics under adversarial training rather than relying on a deterministic autoencoding pathway.

In addition to the adversarial objective, the Generator is regularized by a sample-level proximity term during training, which constrains generated outputs to remain close to the observed input pattern. As a result, the Generator serves as a stochastic sequence model with sample-level consistency regularization. During inference, STGAD draws multiple

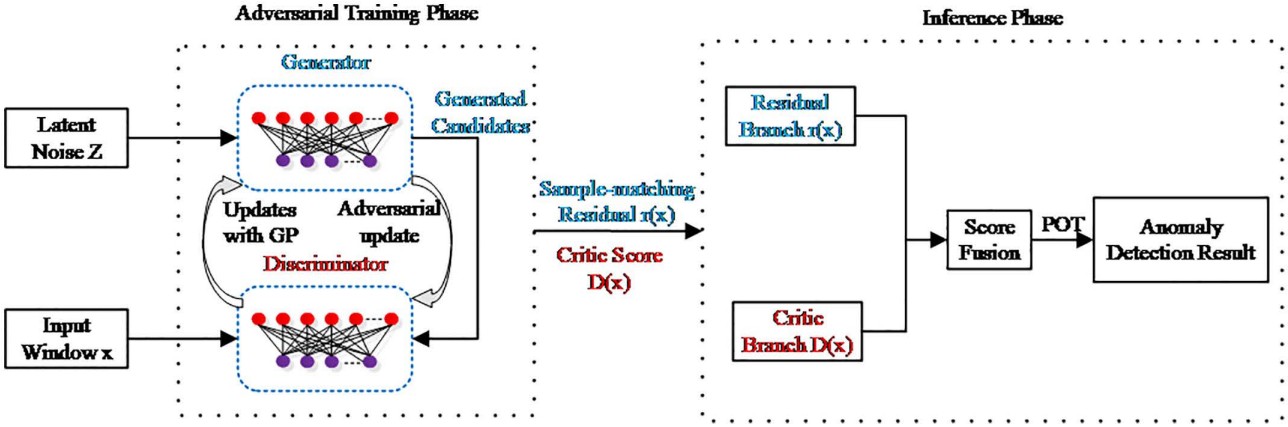

**Fig 1. Overall architecture of the proposed STGAD framework.**

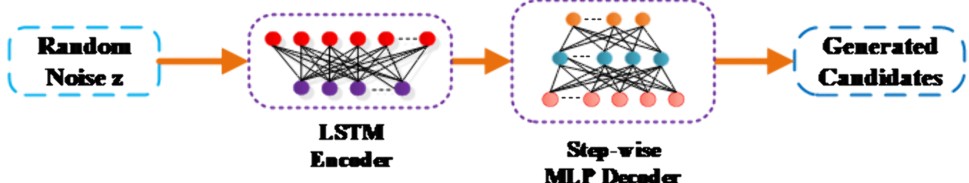

**Fig 2. Internal architecture of the Generator in STGAD.**

generated candidates from noise and uses the minimum residual with respect to the input window as a sample-matching anomaly cue. This residual signal complements the critic-based score from the Discriminator in the final decision process.

Discriminator Architecture. The Discriminator evaluates whether an input window is more consistent with the observed data distribution or with generated samples. As shown in Fig 3, each time step is first projected into a latent representation through a shared linear layer followed by LeakyReLU activation. Sinusoidal positional encodings are then added to preserve temporal order. The resulting sequence is processed by a standard Transformer encoder embedded in the critic, which performs global within-window temporal reasoning through multi-head self-attention. This design enables the Discriminator to capture long-range temporal dependencies and cross-variable interactions before the sequence is summarized by temporal average pooling and mapped to a scalar normality score through a two-layer MLP.

In summary, STGAD combines a stochastic Generator with sample-level proximity regularization and a Transformer-based critic under WGAN-GP training, producing complementary residual-based and critic-based anomaly evidence for downstream detection and localization.

### 3.3. Training and inference

STGAD adopts the Wasserstein GAN with Gradient Penalty (WGAN-GP) to improve adversarial stability and mitigate common failure modes such as mode collapse. During training, the Generator and Discriminator are jointly optimized so that the Generator learns plausible temporal windows under adversarial supervision and sample-level proximity regularization, while the Discriminator learns a stable critic for distinguishing observed from generated sequences. During inference, anomaly likelihood is evaluated using two complementary signals: a sample-matching residual obtained from multiple generated candidates and a critic-based score from the Discriminator.

The Discriminator loss combines the score difference between real and generated sequences with a gradient penalty computed from interpolated samples:

$$\mathcal{L}_D = \mathbb{E}_{\hat{x} \sim P_g} \left[ D(\hat{x}) \right] - \mathbb{E}_{x \sim P_r} \left[ D(x) \right] + \lambda \, \mathbb{E}_{\tilde{x} \sim P_{\tilde{x}}} \left( \| \, \nabla_{\tilde{x}} D\left( \tilde{x} \right) \, \|_2 - 1 \right)^2 \tag{3}$$

The Generator loss is defined as the negative of the Discriminator score on generated samples together with an $L_2$ proximity term:

$$\mathcal{L}_G = -\mathbb{E}_{z \sim P_z} \left[ D\left( G\left( z \right) \right) \right] + \gamma \, \| G\left( z \right) - x \|_2^2 \tag{4}$$

In Eq. (4), the first term encourages the Generator to produce samples that obtain higher scores from the Discriminator, while the second term introduces a sample-level $L_2$ proximity constraint between the generated sequence $G\left( z \right)$ and the

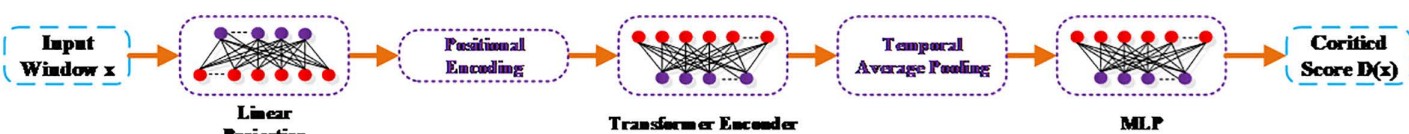

**Fig 3. Internal architecture of the Discriminator in STGAD.**

input window x. This term regularizes the generated trajectory, suppresses excessive deviations, and improves the stability of adversarial optimization.

The overall optimization routine follows the standard WGAN-GP schedule in which the Discriminator is updated multiple times per Generator step. The resulting procedure is summarized in Algorithm 1 (Training Procedure of STGAD), which instantiates the losses in Eqs. (3)–(4) and specifies the update loop.

## Algorithm 1. STGAD Training Algorithm (WGAN-GP)

**Require:** Generator $G$, Discriminator $D$; training window set $W$; batch size $B$; critic steps $n_{critic}$; learning rate $\eta$; gradient-penalty weight $\lambda$; proximity weight $\gamma$; total iterations $T$

1: Initialize weights of $G$ and $D$
2: **for** $t=1$ **to** $T$ **do**
3: **for** $j=1$ **to** $n_{critic}$ **do**
4: Sample real batch $\{x_b\}_{b=1}^{B} \subset W$ and noise $\{Z_b\}_{b=1}^{B} \sim N(0,1)$
5: $\hat{x}_b \leftarrow G(Z_b), \quad b = 1, \dots, B$
6: $\tilde{x}_b \leftarrow \epsilon_b x_b + (1 - \epsilon_b) \hat{x}_b, \quad \epsilon_b \sim U(0, 1)$
7: $L_D \leftarrow \frac{1}{B} \sum_{b=1}^{B} D(\hat{x}_b) - \frac{1}{B} \sum_{b=1}^{B} D(x_b)$
8: $+ \lambda \frac{1}{B} \sum_{b=1}^{B} (\| \nabla_{\tilde{x}_b} D(\tilde{x}_b) \|_2 - 1)^2$
9: Update $D$ using Adam on $\nabla L_D$ with step size $\eta$
10: **end for**
11: Sample real batch $\{x_b\}_{b=1}^{B} \subset W$ and noise $\{Z_b\}_{b=1}^{B} \sim N(0,1)$
12: $\hat{x}_b \longleftarrow G(Z_b), \quad b = 1, \dots, B$
13: $L_G \leftarrow -\frac{1}{B} \sum_{b=1}^{B} D(\hat{x}_b) + \gamma \frac{1}{B} \sum_{b=1}^{B} \| \hat{x}_b - x_b \|_2^2$
14: Update $G$ using Adam on $\nabla L_D$ with step size $\eta$
15: **end for**

During inference, STGAD evaluates anomaly likelihood using two complementary signals. First, the Generator produces multiple candidate windows from independent noise draws, and the minimum residual with respect to the observed input window is recorded:

$$S_{res}(x) = \min_{z_i} \| x - G(z_i) \|^2$$

(5)

Because the Generator is stochastic, multiple candidates are sampled for each window and the minimum residual is retained as a robust sample-matching score. This score approximates the distance between the observed window and the learned normal temporal manifold, while reducing sensitivity to occasional poor generated samples.

Second, the critic-based score is computed as:

$$S_{critic}(x) = 1 - D(x)$$

(6)

The final anomaly score is obtained by fusing the normalized residual-based and critic-based components:

$$S(x) = \beta \cdot S_{res}(x) + (1 - \beta) \cdot S_{critic}(x)$$

(7)

In practice, both per-window scores are min–max normalized on the validation split before fusion. The fusion weight β is selected through validation sensitivity analysis and then fixed during testing. Final anomaly decisions are produced using POT-based thresholding. The inference and dual-score anomaly scoring procedure is summarized in Algorithm 2.

**Algorithm 2 STGAD Inference and Dual-Score Anomaly Scoring**

```
Require: Trained G,D; test window x; number of draws K; fusion weight β ∈ [0,1]; validation statis-
tics for min-max normalization; threshold τ
1:  for k=1 to K do
2:      Sample z_k ∼ N(0,1)
3:      x̂^(k) ← G(z_k)
4:      r_k ←‖ x − x̂^(k) ‖ 1
5:  end for
6:  r(x) ← min_{1≤k≤K}r_k   ▷(sample-matching residual)
7:  (x) ← −D(x)   ▷ (critic-based score)
8:  Normalize r(x) and s_critic(x) to [0,1] using validation statistics
9:  Obtain normalized scores r̄(x) and s̄_critic(x)
10: S(x) ← βr̄(x) + (1 − β)s̄_critic(x)
11: if S(x) ≥ τ then
12:     Label x as anomalous
13: else
14:     Label x as normal
15: end if
16: return S(x) and the predicted label
```

Both the Generator and Discriminator are optimized using the Adam optimizer (learning rate = 0.001, $\beta_1$ = 0.5, $\beta_2$ = 0.9), with batch size 64. The time-series data is segmented into overlapping windows and reshaped into $L \times d$ sequences, where $L$ is the number of time steps per window and $d$ is the number of variables. Gradient clipping and early stopping are applied to ensure training stability and generalization. The model is implemented in PyTorch and supports GPU acceleration.

## 4. Experimental setup

This section details the experimental protocol used to assess STGAD's effectiveness. Presented first are the benchmark datasets together with their preprocessing procedures; next, we specify the windowing scheme and model hyperparameters. To conclude, we lay out the evaluation metrics and the baseline comparison configuration so that results remain fair and reproducible.

### 4.1. Datasets and preprocessing

We assess STGAD on five widely used multivariate time-series anomaly benchmarks: SMD, SMAP, MSL, SWaT, and MIT-BIH. These corpora cover varied application areas—cloud operations, industrial control, aerospace telemetry, and physiological signal analysis. Brief summaries are provided below:

SMD (Server Machine Dataset) [43]: Multivariate system metrics from 28 cloud servers, each with synthetic anomalies. Features include CPU, memory, and network load.

SMAP & MSL (NASA Telemetry) [44]: Satellite and rover telemetry data with labeled anomalies due to system faults. Data contains multiple correlated physical channels.

SWaT [45]: Cyber-physical datasets collected from industrial control system testbeds. Each includes attack scenarios labeled as anomalies across water treatment and distribution systems.

MIT-BIH (MIT-BIH Arrhythmia) [46]: ECG recordings annotated with cardiac arrhythmias, converted into multivariate time series to test sensitivity to periodic yet irregular biological signals.

All datasets are standardized with per-variable z-score normalization computed exclusively from the training split; the resulting means and standard deviations are then applied unchanged to validation and test data. When an official train/test split is available, we adopt it verbatim; otherwise, we use a chronological split in which the first 80% of each series constitutes the training portion (assumed normal) and the remaining 20% serves as the test set. The anomaly labels in our

benchmarks arise either from naturally occurring events or from controlled synthetic injections designed to probe robustness; together, these cases cover both realistic operational drifts and stress scenarios.

For window-level evaluation, we follow the point-adjusted convention used in our evaluation section: a window is marked anomalous if any timestamp within the window overlaps a ground-truth anomaly segment. To ensure reproducibility, all normalization statistics are derived solely from the training split, no resampling is performed, and each dataset's native sampling rate and channel configuration are preserved throughout preprocessing and evaluation.

## 4.2. Windowing and model configuration

Fig 4 summarizes the end-to-end procedure used in our experiments, from preprocessing to inference and score fusion. We segment each multivariate time series into overlapping windows, with window length $L = 5$ and stride $s = 1$. This dense windowing scheme increases the effective number of training samples and allows the model to observe short-period regularities and typical lead–lag relationships across variables in a consistent manner.

We choose $L = 5$ as the default setting because short-to-moderate windows provide the best balance between temporal context and anomaly localization. In our setting, shorter windows preserve localized anomaly signatures and reduce the risk that weak or brief abnormal patterns are diluted by excessive surrounding context. At the same time, they keep the within-window variability and computational cost manageable. This choice is also supported by the sensitivity analysis in Section 5.4, where short-to-moderate window lengths consistently provide a strong accuracy–efficiency trade-off across datasets. Unless otherwise stated, the same window length and stride are used for all datasets to maintain a unified experimental protocol.

During inference, the residual-based score and the critic-based score are min–max normalized using statistics computed on the validation split; these normalization statistics remain fixed at test time. The two normalized scores are then fused with equal weighting, i.e., $\beta = 0.5$ for the residual-based component and $1 - \beta = 0.5$ for the critic-based component.

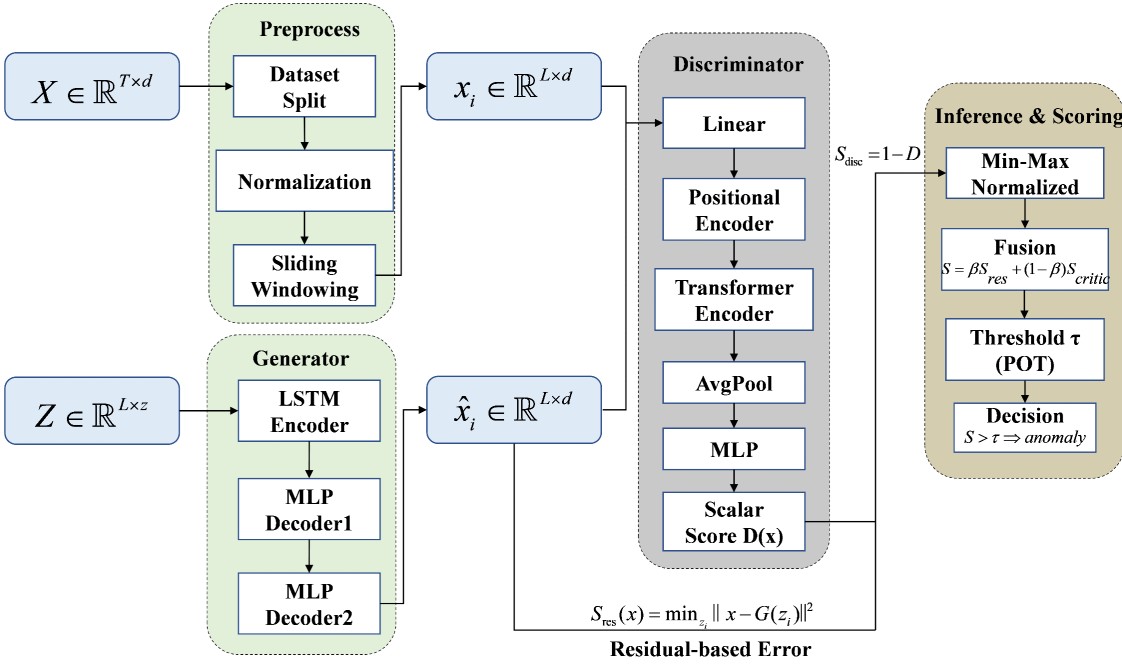

**Fig 4. Windowing and fusion procedure.**

The fusion weight is selected based on validation sensitivity analysis and kept unchanged during testing. After fusion, the decision threshold is determined on the validation split using POT and then applied unchanged to the test split to produce timeline-level anomaly decisions.

All remaining hyperparameters are kept fixed unless a dataset imposes a documented domain-specific constraint. We train the model using Adam with a learning rate of 0.001 and a batch size of 64, together with gradient clipping and early stopping based on validation performance. Implementation details of adversarial optimization and update scheduling are reported in Section 3 and are not repeated here.

### 4.3. Evaluation protocol

We assess STGAD and all baselines with standard anomaly-detection criteria—Precision, Recall, F1-score, and ROC-AUC. A point-adjusted protocol is employed: an anomalous episode counts as detected if any timestamp within its span is flagged. For datasets with pronounced class imbalance, we additionally report Precision–Recall AUC (PR-AUC) or a Composite F1-score. To bolster reliability, each metric is averaged across multiple independent runs.

For fair benchmarking, we evaluate all baseline methods—DAGMM, USAD, OmniAnomaly, MAD-GAN, and Tra-nAD—in an identical unsupervised setting. This means all models are trained solely on normal data, employing identical windowing preprocessing and applying uniform scoring rules consistent with STGAD. Furthermore, we use each baseline's original open-source implementation with its default hyperparameter settings (as reported in the respective papers), without any additional tuning. Thresholding is applied uniformly to all methods. For each model, we fit the POT threshold on validation anomaly scores and then apply the resulting threshold unchanged to the test split. This protocol avoids using test labels for threshold selection and ensures a consistent, fair comparison across STGAD and all baselines.

## 5. Results and analysis

This section summarizes the empirical results of STGAD. We first present benchmark performance and efficiency comparisons, followed by ablation and stability analyses. We then report robustness, sensitivity, and interpretability results to further characterize the model's behavior in practical settings.

### 5.1. Comparative evaluation with baseline methods

To evaluate the effectiveness of the proposed STGAD model comprehensively, we compare its performance with five representative baseline methods: TranAD, USAD, OmniAnomaly, DAGMM, and MAD-GAN. Experiments are conducted on five benchmark multivariate time-series anomaly detection datasets: MIT-BIH, SMAP, MSL, SWaT, and SMD. We report four standard metrics—Precision (P), Recall (R), F1-score, and Area Under the ROC Curve (AUC)—to provide a comprehensive assessment of detection accuracy and robustness. The performance comparison across the five datasets is summarized in Table 2.

As summarized in Table 2, STGAD achieves the best or near-best results across the five datasets, with consistently strong F1 and AUC. The improvements are most evident on MIT-BIH, MSL, and SMD, while on SWaT (highly imbalanced and noisy), all methods exhibit lower recall and STGAD remains competitive.

In addition to detection accuracy, we report computational cost to complement the practical evaluation in Tables 3 and 4.

Table 3 summarizes the wall-clock training time of each method for five epochs under the same experimental environment. Overall, STGAD exhibits competitive training efficiency relative to several reconstruction-based and probabilistic baselines, while remaining substantially faster than heavier generative models such as OmniAnomaly on larger datasets.

Table 4 reports the inference throughput and latency of STGAD on an RTX 4060 Laptop GPU under the default sampling setting used in our experiments. Across datasets, the model processes approximately 793–861 windows per second, corresponding to about 1.16–1.26 ms per window. These results indicate that STGAD is suitable for offline analysis

**Table 2. Performance comparison of STGAD and baseline methods across multiple datasets.**

| Method | MIT-BIH | | | | SMAP | | | | MSL | | | |
|---|---|---|---|---|---|---|---|---|---|---|---|---|
| | P | R | AUC | F1 | P | R | AUC | F1 | P | R | AUC | F1 |
| STGAD | 0.9423 | 1.0000 | 0.9843 | 0.9703 | 0.8358 | 1.0000 | 0.9905 | 0.9105 | 0.9226 | 1.0000 | 0.9933 | 0.9597 |
| TranAD | 0.8667 | 1.0000 | 0.9606 | 0.9286 | 0.8043 | 1.0000 | 0.9883 | 0.8915 | 0.8908 | 1.0000 | 0.9902 | 0.9422 |
| USAD | 0.8938 | 1.0000 | 0.9696 | 0.9439 | 0.7480 | 0.9627 | 0.9890 | 0.8419 | 0.7949 | 0.9912 | 0.9795 | 0.8822 |
| OA | 0.8541 | 1.0000 | 0.9563 | 0.9213 | 0.8130 | 0.9419 | 0.9889 | 0.8728 | 0.7848 | 0.9924 | 0.9782 | 0.8765 |
| DAGMM | 0.9282 | 1.0000 | 0.9802 | 0.9628 | 0.8069 | 0.9891 | 0.9885 | 0.8888 | 0.7363 | 1.0000 | 0.9532 | 0.7721 |
| MAD-GAN | 0.9396 | 1.0000 | 0.9836 | 0.9689 | 0.8157 | 0.9216 | 0.9891 | 0.8654 | 0.8516 | 0.9930 | 0.9862 | 0.9169 |

| Method | SWaT | | | | SMD | | | | | | | |
|---|---|---|---|---|---|---|---|---|---|---|---|---|
| | P | R | AUC | F1 | P | R | AUC | F1 | | | | |
| STGAD | 1.0000 | 0.6879 | 0.8439 | 0.8151 | 0.9548 | 0.9974 | 0.9962 | 0.9756 | | | | |
| TranAD | 0.9977 | 0.6879 | 0.8438 | 0.8143 | 0.9017 | 0.9974 | 0.9931 | 0.9471 | | | | |
| USAD | 0.9977 | 0.6879 | 0.8460 | 0.8143 | 0.9060 | 0.9974 | 0.9933 | 0.9495 | | | | |
| OA | 0.9782 | 0.6957 | 0.8467 | 0.8131 | 0.8881 | 0.9985 | 0.9946 | 0.9401 | | | | |
| DAGMM | 0.7778 | 0.5109 | 0.7140 | 0.6167 | 0.9103 | 0.9914 | 0.9954 | 0.9491 | | | | |
| MAD-GAN | 0.9593 | 0.6957 | 0.8463 | 0.8065 | 0.9991 | 0.8440 | 0.9933 | 0.9150 | | | | |

**Table 3. Training time (seconds) for 5 epochs.**

| Method | MIT-BIH | SMAP | MSL | SWaT | SMD |
|---|---|---|---|---|---|
| TranAD | 3.31 | 1.7878 | 2.5433 | 1.2494 | 64.58 |
| USAD | 158.7581 | 54.97 | 34.6419 | 53.8327 | 477.0034 |
| DAGMM | 119.9868 | 39.1755 | 26.5745 | 53.4132 | 390.9106 |
| OmniAnomaly | 477.7496 | 180.3296 | 113.0611 | 176.3227 | 1930.6114 |
| MAD-GAN | 159.2908 | 82.8039 | 122.4641 | 54.9408 | 584.0891 |
| STGAD | 37.003 | 14.040 | 10.327 | 14.230 | 152.851 |

**Table 4. STGAD inference throughput and latency.**

| Dataset | Throughput (win/s) | Latency (ms/win) | Peak alloc (GB) | Peak reserved (GB) |
|---|---|---|---|---|
| MIT-BIH | 857.7 | 1.166 | 0.024 | 0.047 |
| SMAP | 793.3 | 1.261 | 0.024 | 0.047 |
| MSL | 825.2 | 1.212 | 0.024 | 0.047 |
| SWaT | 861.3 | 1.161 | 0.024 | 0.047 |
| SMD | 824.5 | 1.213 | 0.024 | 0.047 |

and near-real-time monitoring scenarios. Since the residual branch relies on sample matching over multiple generated candidates, the inference cost increases approximately linearly with the sampling budget N. In practice, this means that a moderate N offers a favorable balance between residual robustness and latency, while stricter real-time deployments may reduce N to satisfy tighter timing constraints.

## 5.2. Ablation analysis

To further evaluate the contribution of each component in STGAD, we conduct an ablation study focusing on three key design factors: the attention-based temporal interaction module in the critic, the Transformer encoder used for

within-window temporal modeling, and the WGAN-GP training objective. Accordingly, we construct three variant models for comparison.

w/o Self-Attn: removes the multi-head self-attention operation from the critic while retaining the remaining projection and scoring layers.

w/o Transformer: replaces the Transformer encoder in the critic with a simplified single-layer MLP, thereby removing structured within-window temporal modeling.

w/o WGAN-GP: replaces the WGAN-GP objective with the standard GAN loss, in order to assess the effect of Wasserstein training with gradient penalty on optimization stability and detection performance.

For brevity, Table 5 uses the abbreviations w/o Attn for w/o Self-Attn, w/o Trans for w/o Transformer, and VGAN for the variant without WGAN-GP. Each variant is evaluated across the five benchmark datasets (MIT-BIH, SMAP, MSL, SWaT, and SMD). Table 5 presents the comparative results across Precision (P), Recall (R), ROC-AUC, and F1-score metrics.

Table 5 shows that removing either attention or the temporal encoder reduces performance, confirming that both cross-time modeling and within-window temporal encoding are beneficial.

The effect of removing the temporal encoder is dataset-dependent and is particularly pronounced on SMD, where F1 drops substantially, indicating that richer temporal encoding is important for complex telemetry.

Replacing WGAN-GP with a vanilla GAN objective (VGAN) leads to severe performance degradation on multiple datasets, often collapsing to near-zero F1, highlighting the importance of stable adversarial training.

In addition to the endpoint metrics in Table 5, Fig 5 and Table 6 provide training-stability evidence for the effect of WGAN-GP. Under the standard GAN objective (STGAD w/o WGAN-GP), the discriminator shows clear saturation behavior, whereas the WGAN-GP variant maintains a well-controlled gradient penalty throughout training.

Consistently, Table 6 shows that WGAN-GP substantially reduces the variability of discriminator loss across datasets, indicating smoother and more stable adversarial optimization. Together, these results support that WGAN-GP is an important component for reliable training in STGAD.

## 5.3. Robustness analysis

To evaluate robustness under sensor noise, we conduct Gaussian noise injection experiments by adding zero-mean Gaussian noise to the input time series during training. We consider two noise intensities, $\sigma = 0.1$ and $\sigma = 0.25$, and evaluate all methods under the same noise settings for a fair comparison on five benchmark datasets (MIT-BIH, SMAP, MSL, SWaT, and SMD). We report AUC and F1-score to assess both ranking quality and detection accuracy under noise in Table 7.

**Table 5. Ablation study comparing STGAD and its variant models on multiple datasets.**

| Model | MIT-BIH | | | | SMAP | | | | MSL | | | |
|---|---|---|---|---|---|---|---|---|---|---|---|---|
| | P | R | AUC | F1 | P | R | AUC | F1 | P | R | AUC | F1 |
| w/o Attn | 0.8611 | 0.9231 | 0.9234 | 0.8910 | 0.8339 | 1.0000 | 0.9904 | 0.9094 | 0.9064 | 1.0000 | 0.9918 | 0.9509 |
| w/o Trans | 0.9506 | 1.0000 | 0.9867 | 0.9747 | 0.8348 | 1.0000 | 0.9904 | 0.9100 | 0.9172 | 1.0000 | 0.9928 | 0.9568 |
| VGAN | 0.8644 | 1.0000 | 0.9598 | 0.9272 | 0.0000 | 0.0000 | 0.0000 | 0.0000 | 0.0000 | 0.0000 | 0.0000 | 0.0000 |
| **STGAD** | **0.9423** | **1.0000** | **0.9843** | **0.9703** | **0.8358** | **1.0000** | **0.9905** | **0.9105** | **0.9226** | **1.0000** | **0.9933** | **0.9597** |

| Model | SWaT | | | | SMD | | | |
|---|---|---|---|---|---|---|---|---|
| | P | R | AUC | F1 | P | R | AUC | F1 |
| w/o Attn | 0.1000 | 0.7391 | 0.3776 | 0.1762 | 0.9398 | 0.7896 | 0.8922 | 0.8582 |
| w/o Trans | 0.9955 | 0.6879 | 0.8437 | 0.8136 | 0.9427 | 0.8260 | 0.9104 | 0.8805 |
| VGAN | 0.0000 | 0.0000 | 0.0000 | 0.0000 | 0.0000 | 0.0000 | 0.0000 | 0.0000 |
| **STGAD** | **1.0000** | **0.6879** | **0.8439** | **0.8151** | **0.9548** | **0.9974** | **0.9962** | **0.9756** |

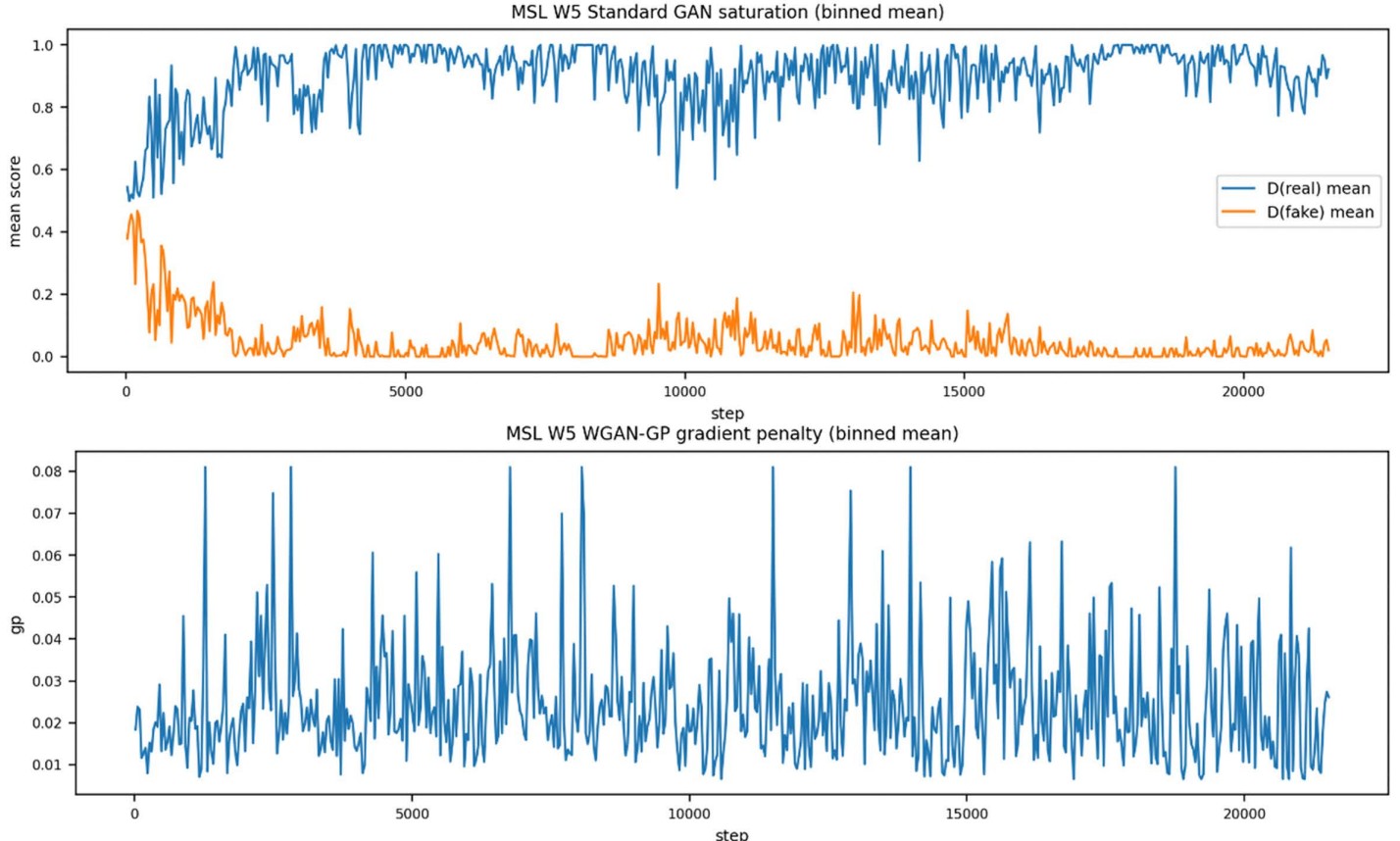

**Fig 5. Representative training stability evidence on MSL.**

**Table 6. Training-loss variability (std) under standard GAN vs WGAN-GP.**

| Datasets | d_lossstd (VGAN) | d_lossstd (STGAD) | g_lossstd (VGAN) | g_lossstd (STGAD) |
|---|---|---|---|---|
| MSL | 3.773 | 0.920 | 11.04 | 0.960 |
| SMAP | 3.319 | 3.009 | 9.786 | 1.916 |
| SWaT | 0.768 | 0.196 | 2.147 | 4.767 |
| MIT-BIH | 1.847 | 0.351 | 3.848 | 3.559 |
| SMD | 2.275 | 1.402 | 8.674 | 1.951 |

While most methods perform well at $\sigma = 0.1$, several baselines degrade noticeably when noise increases to $\sigma = 0.25$. In contrast, STGAD shows comparatively stable performance across the two noise levels on all datasets, maintaining strong AUC and F1. Overall, the robustness of STGAD is consistent with its design of combining complementary signals and modeling temporal structure within windows, which reduces sensitivity to noise perturbations.

## 5.4. Sensitivity analysis

We investigate the sensitivity of STGAD to three key hyper-parameters that affect the final detection behavior and efficiency: (i) the fusion weight $\beta$ for combining critic-based and residual-based anomaly cues, (ii) the number of generator

**Table 7. Robustness Comparison under Gaussian Noise (AUC/ F1).**

| Method | MIT-BIH | | SMAP | | MSL | | SWaT | | SMD | |
|---|---|---|---|---|---|---|---|---|---|---|
| | AUC | F1 | AUC | F1 | AUC | F1 | AUC | F1 | AUC | F1 |
| TranAD (0.1) | 0.99 | 0.99 | 0.99 | 0.88 | 0.99 | 0.90 | 0.84 | 0.82 | 1.00 | 0.98 |
| TranAD (0.25) | 0.50 | 0.00 | 0.70 | 0.57 | 0.99 | 0.95 | 0.66 | 0.49 | 0.60 | 0.34 |
| DAGMM (0.1) | 0.99 | 0.99 | 0.99 | 0.91 | 0.98 | 0.90 | 0.84 | 0.81 | 1.00 | 1.00 |
| DAGMM (0.25) | 0.52 | 0.09 | 0.70 | 0.57 | 0.98 | 0.90 | 0.50 | 0.00 | 0.60 | 0.34 |
| OmniAnomaly (0.1) | 1.00 | 1.00 | 0.99 | 0.92 | 0.99 | 0.95 | 0.84 | 0.81 | 0.92 | 0.91 |
| OmniAnomaly (0.25) | 0.95 | 0.91 | 0.70 | 0.57 | 0.99 | 0.95 | 0.50 | 0.00 | 0.60 | 0.34 |
| MAD-GAN (0.1) | 0.95 | 0.94 | 0.99 | 0.92 | 0.98 | 0.90 | 0.84 | 0.91 | 0.82 | 0.79 |
| MAD-GAN (0.25) | 0.50 | 0.00 | 0.70 | 0.58 | 0.98 | 0.91 | 0.50 | 0.00 | 0.60 | 0.34 |
| **STGAD (0.1)** | **0.99** | **0.98** | **0.99** | **0.91** | **0.99** | **0.96** | **0.84** | **0.81** | **1.00** | **0.98** |
| **STGAD (0.25)** | **0.94** | **0.92** | **0.99** | **0.91** | **0.99** | **0.95** | **0.84** | **0.81** | **0.99** | **0.92** |

samples N used to compute the sample-matching residual, and (iii) the window length L that determines the temporal context seen by the temporal encoder and discriminator. Unless otherwise specified, we vary one factor at a time while keeping the remaining settings fixed, and report the F1-score trends across datasets. The corresponding results are summarized in Fig 6.

Overall, STGAD shows stable performance over a broad range of configurations, and the trends are consistent across datasets. First, the fusion weight β mainly controls the relative reliance on critic evidence versus sample-matching residuals. We observe that extremely small β is more likely to underperform, while mid-to-high β typically yields strong and stable results. This indicates that the residual term provides a reliable complementary signal, and combining the two cues is generally more robust than relying on either alone.

Second, increasing the number of generator samples N helps mitigate generation stochasticity and reduces the chance that the residual-based cue is dominated by occasional poor samples. In practice, performance usually improves when moving from very small N to a moderate range, after which the gains gradually saturate. Since the computational cost grows approximately linearly with N, the choice of N should be matched to the deployment budget. For offline analysis and post-event diagnosis, a moderate sampling budget is preferable because it provides a more stable residual estimate and stronger detection robustness. For near-real-time monitoring, the default setting offers a practical balance between residual stability and latency. For stricter real-time scenarios, N can be reduced to meet tighter latency constraints, with

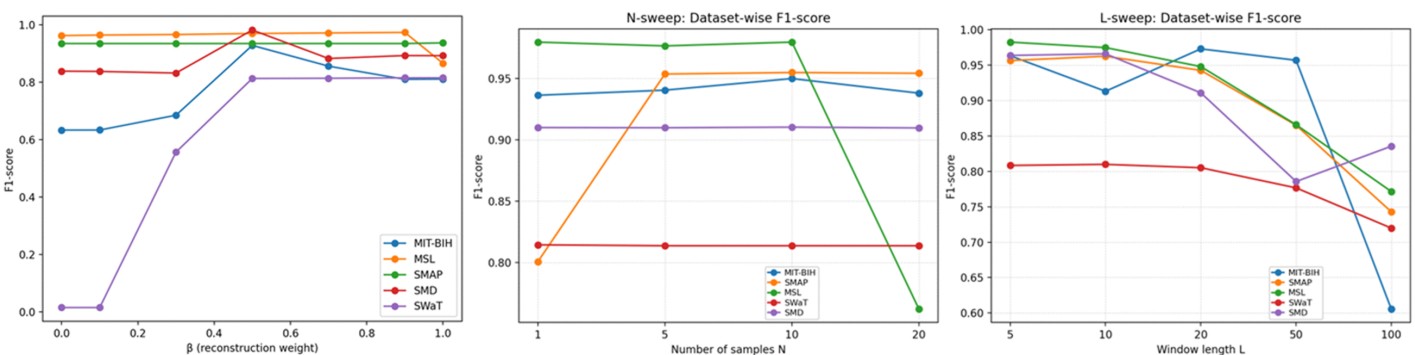

**Fig 6. Sensitivity analysis of STGAD to key hyperparameters (β, N, and window length L).**

the understanding that this may slightly weaken the robustness of the sample-matching residual. Therefore, N serves as a controllable knob for trading residual stability against inference cost under different operational requirements.

Third, the window length L determines how much temporal context is available for within-window modeling. We find that short-to-moderate window lengths are generally preferable: they preserve localized anomaly signatures while keeping variability and computational cost manageable. In contrast, overly long windows may dilute short abnormal patterns with excessive surrounding context and can introduce unnecessary complexity, which may reduce both detection effectiveness and efficiency.

Based on these observations, we adopt $\beta = 0.5$, $N = 10$, and $L = 5$ as the default configuration. In particular, the choice $L = 5$ reflects a practical balance: it preserves localized anomaly signatures, avoids unnecessary context dilution, and consistently provides a strong accuracy–efficiency trade-off across datasets under a unified configuration.

### 5.5. Interpretability analysis

We assess interpretability primarily on SMAP, a NASA telemetry dataset with strong cross-channel coupling and expert-labeled drift and step anomalies. All analyses are performed on fixed-length windows under the same training-split z-score normalization used in the main experiments. For visualization, each anomalous window is paired with a nearest-normal reference obtained by averaging its K closest normal windows. Explanations are produced using two complementary views: (i) a temporal attention map that highlights influential time steps, and (ii) gradient-based feature saliency that ranks influential variables.

To relate model saliency to model-agnostic statistical evidence, we compute two simple feature rankings for each window: (i) Max $|\Delta|$, the largest within-window change for each variable, and (ii) Max $|z|$, the largest standardized deviation for each variable. Agreement is summarized by overlap@5 between the saliency Top-5 variables and the Top-5 variables ranked by these two statistics. We refer to these measures as Saliency–Local overlap@5 (vs. Max $|\Delta|$) and Saliency–Global overlap@5 (vs. Max $|z|$), respectively. In addition, we summarize decision strength using the score margin (score minus threshold) and the score percentile among all windows.

We illustrate interpretability on a true-positive SMAP window by examining attribution along both the temporal and variable axes. As shown in Fig 7A, a small subset of channels dominates the saliency ranking. Several of these variables also appear among the Top-K features ranked by Max $|\Delta|$ and Max $|z|$ in Fig 7B, indicating that the model's attribution is aligned with two intuitive model-agnostic anomaly cues: abrupt local changes and globally rare deviations.

In Fig 8, we overlay the anomalous window against the nearest-normal reference for the Top-6 salient channels and annotate each subplot with a confidence ribbon. Several structured departures can be observed, including persistent step-like drops, pulse-shaped excursions, and late rebounds that are absent in the normal reference windows. These deviations co-occur with attribution peaks in Fig 7A and with high rankings under Max $|\Delta|$ or Max $|z|$ in Fig 7B, providing channel-level evidence that is broadly consistent with the saliency interpretation.

While the SMAP case study provides an intuitive visualization, interpretability should also be examined at the distribution level. We therefore quantify agreement between saliency-ranked variables and statistical cues using overlap@5 over quantile-sampled true-positive windows. Specifically, true-positive windows are defined under the same evaluation protocol as the main experiments, stratified by confidence margin, and sampled across multiple margin quantiles. Table 8 and Table 9 report the median (IQR) overlap values and the number of sampled windows for each dataset.

As shown in Table 8 and Table 9, Saliency–Local overlap@5 is consistently high on several datasets, suggesting that abrupt local changes are an important cue captured by the model in these domains. In contrast, Saliency–Global overlap@5 varies more across datasets, reflecting differences in how anomalies manifest. For example, the lower overlap values on SMD suggest that anomalies can be more distributed, weaker, or less attributable to a small set of variables, whereas the near-perfect overlap on SWaT and MIT-BIH indicates that anomalies are often concentrated in a small subset of channels.

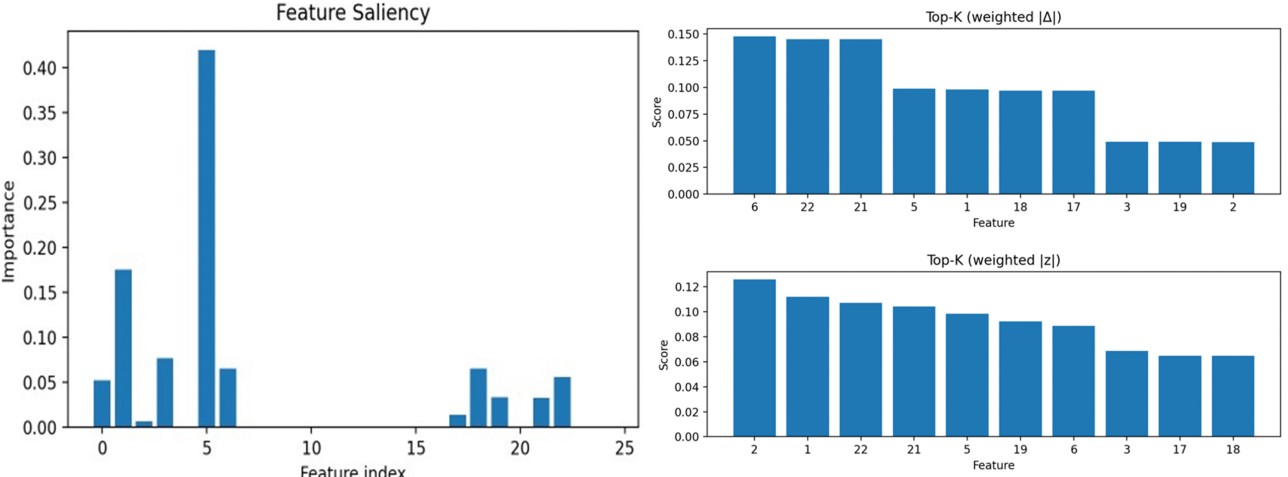

**Fig 7. Feature Attribution and Top-K Statistical Deviations (SMAP).**

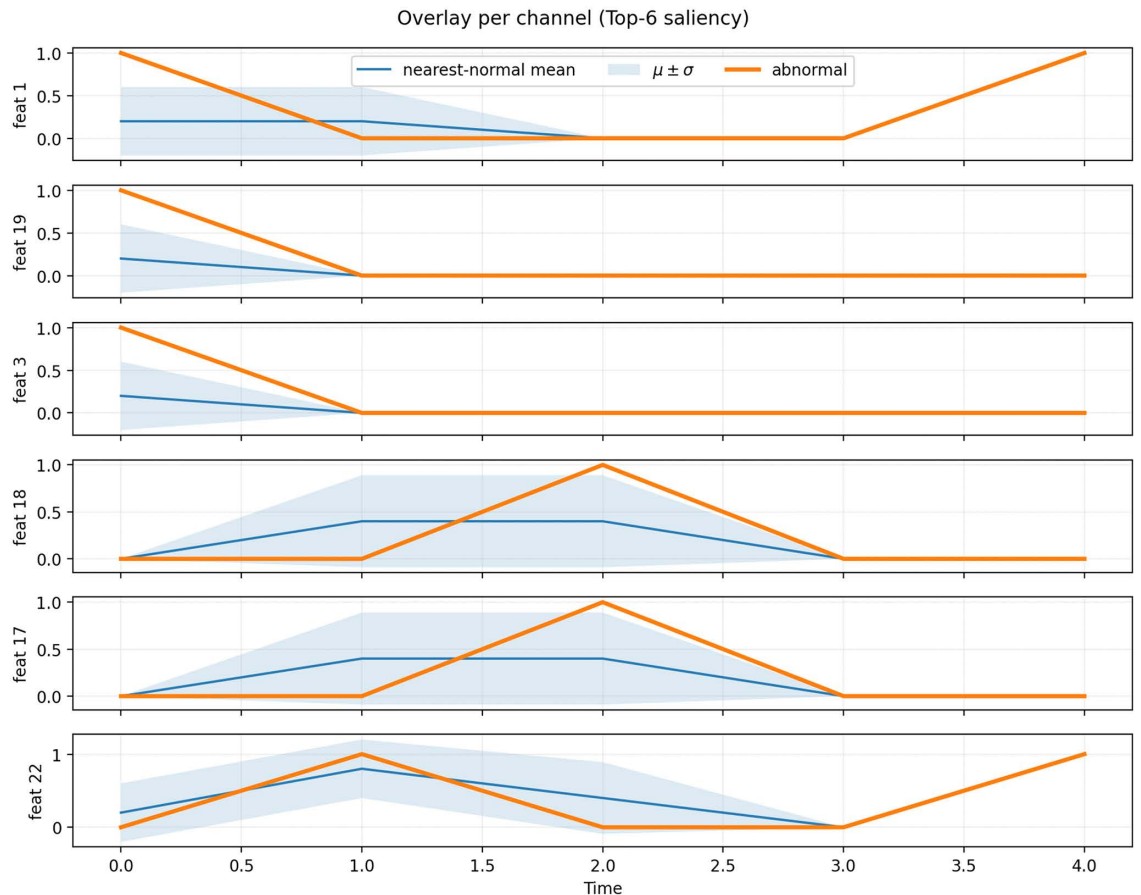

**Fig 8. Channel-wise Shape Deviations on SMAP (Top-6 Salient Variables).**

**Table 8. Saliency–Local overlap@5 (vs. Max|Δ|).**

| Dataset | Q1[0.00,0.33] | Q2[0.33,0.66] | Q3[0.66,1.00] | ALL |
|---------|---------------|---------------|---------------|-----|
| MIT-BIH | 1.00 (0.00), n=20 | 1.00 (0.00), n=20 | 1.00 (0.00), n=20 | 1.00 (0.00), n=60 |
| SMAP | 1.00 (0.00), n=20 | 1.00 (0.00), n=19 | 0.70 (0.60), n=20 | 1.00 (0.20), n=59 |
| MSL | 1.00 (0.00), n=6 | 1.00 (0.00), n=6 | 1.00 (0.00), n=6 | 1.00 (0.00), n=18 |
| SWaT | 1.00 (0.00), n=7 | 1.00 (0.00), n=7 | 1.00 (0.00), n=7 | 1.00 (0.00), n=21 |
| SMD | 0.20 (0.05), n=20 | 0.20 (0.05), n=20 | 0.20 (0.05), n=20 | 0.20 (0.05), n=60 |

**Table 9. Saliency–Global overlap@5 (vs. Max|z|).**

| Dataset | Q1[0.00,0.33] | Q2[0.33,0.66] | Q3[0.66,1.00] | ALL |
|---------|---------------|---------------|---------------|-----|
| MIT-BIH | 1.00 (0.00), n=20 | 1.00 (0.00), n=20 | 1.00 (0.00), n=20 | 1.00 (0.00), n=60 |
| SMAP | 0.60 (0.00), n=20 | 0.60 (0.10), n=19 | 0.60 (0.25), n=20 | 0.60 (0.10), n=59 |
| MSL | 0.90 (0.20), n=6 | 0.60 (0.00), n=6 | 0.70 (0.35), n=6 | 0.80 (0.20), n=18 |
| SWaT | 1.00 (0.00), n=7 | 1.00 (0.00), n=7 | 1.00 (0.00), n=7 | 1.00 (0.00), n=21 |
| SMD | 0.20 (0.05), n=20 | 0.10 (0.20), n=20 | 0.20 (0.05), n=20 | 0.20 (0.20), n=60 |

We emphasize that these explanations should be interpreted as anomaly localization or attribution cues rather than strict root-cause identification. From an operational perspective, these variable-level localization results can be used as diagnostic prioritization cues rather than as fully automatic root-cause decisions. In practice, once an anomalous window is detected, practitioners can first inspect the top-ranked variables and compare them with known sensor-to-subsystem mappings, control loops, or equipment modules. This can help narrow the troubleshooting scope, prioritize targeted sensor validation and subsystem inspection, and support alarm triage in monitoring centers. Therefore, the localization output of STGAD is most useful as a decision-support layer that helps experts focus on the most relevant channels before conducting deeper causal or engineering analysis. Nevertheless, the agreement statistics complement the case-study visualizations and provide a dataset-level view of how consistently the model's saliency aligns with simple statistical evidence.

In Fig 9, we further visualize the alignment between the predicted anomaly score and the ground-truth labels on SMAP using a small subset of representative channels, including highly attributed channels identified in the case study. In these panels, the normalized time series is shown in blue, the model's anomaly score in green, purple shading marks the ground-truth anomalous intervals, and orange shading highlights the predicted anomalous intervals.

The plots reveal prompt, clearly discernible responses at genuine fault onsets: predicted anomalous spans closely track the labeled intervals with minimal localization error, while scores remain low and steady throughout nominal periods—evidence of resilience to transient noise. Taken together, the SMAP case confirms precise timing capture and strong temporal alignment, in line with the quantitative results reported earlier.

## 6. Conclusion

In this paper, we presented STGAD, an unsupervised generative-adversarial framework for multivariate time-series anomaly detection and localization. STGAD integrates a standard Transformer encoder into the critic of a WGAN-GP framework, enabling global within-window temporal modeling under stable adversarial training. This critic-side temporal modeling is complemented by a sample-matching residual derived from a stochastic generator with sample-level proximity regularization, and the two signals are fused for final anomaly scoring.

Experiments on five benchmark datasets—MIT-BIH, SMAP, MSL, SWaT, and SMD—show that STGAD achieves strong and consistent performance across diverse anomaly-detection scenarios. Additional ablation, robustness, sensitivity, and

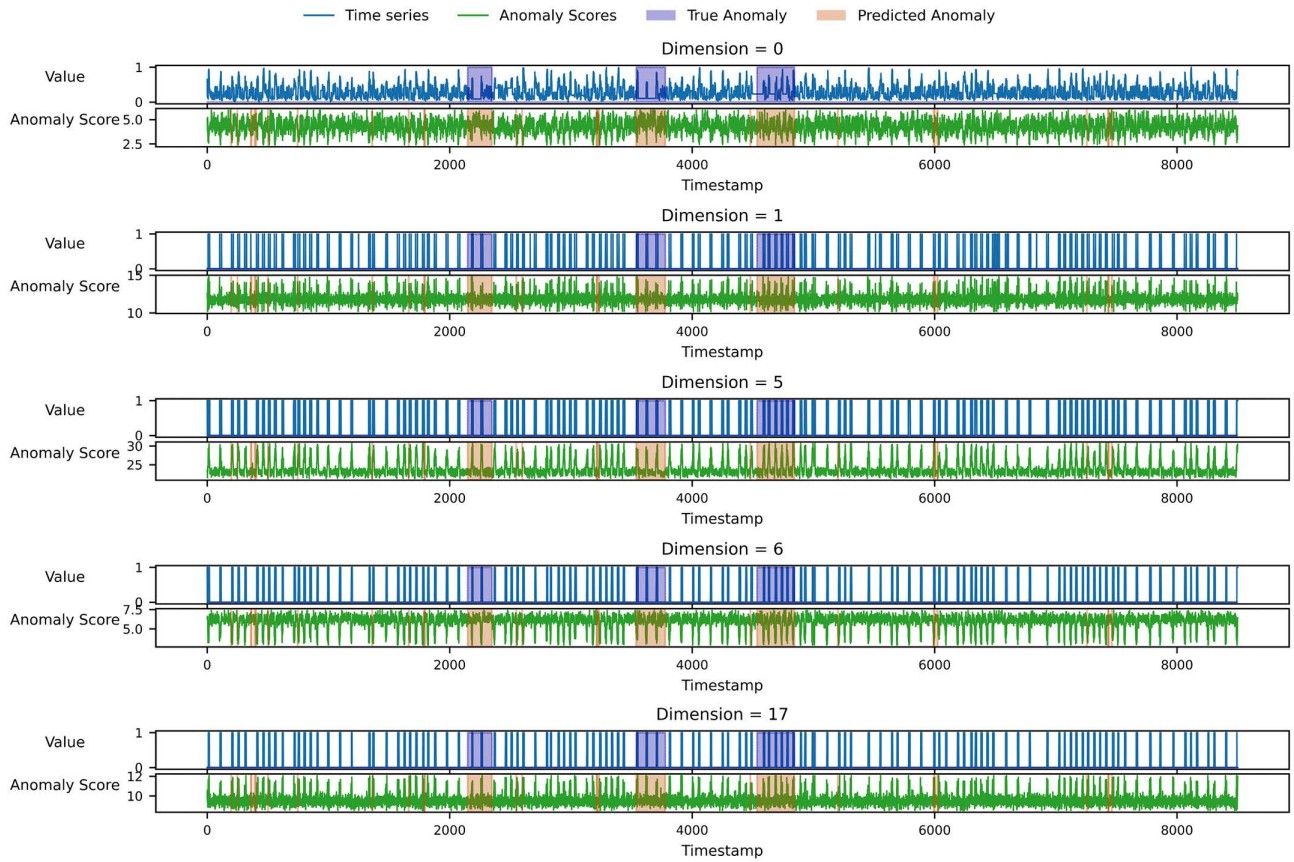

**Fig 9. Qualitative Alignment of Predicted and True Anomalies on SMAP.**

interpretability analyses further support the effectiveness of the overall design, particularly the combination of critic-side temporal modeling, dual-score fusion, and stable adversarial optimization.

From a practical perspective, the results suggest that STGAD provides a favorable balance between detection effectiveness and deployment cost, especially for offline analysis and near-real-time monitoring settings. At the same time, the sensitivity analysis indicates that moderate sampling and short-to-moderate windows are sufficient to capture most of the performance gains without incurring excessive computational overhead. Moreover, the variable-level localization output can serve as a practical prioritization signal for downstream diagnosis, while the sampling budget N can be adjusted according to the latency requirements of different deployment scenarios.

Future work may extend STGAD toward online and incremental learning, multi-resolution temporal modeling, and privacy-preserving or federated deployments. These directions may further improve the applicability of the framework in large-scale cyber-physical monitoring environments.Abbreviations

## Author contributions

**Conceptualization:** Xiao Liao, Yihan Mu.

**Data curation:** Xiao Liao.

**Formal analysis:** Xiao Liao, Yihan Mu.

**Funding acquisition:** Wei Deng, Hongyue Ma.

**Investigation:** Xiao Liao, Wei Deng, Yihan Mu.

**Methodology:** Xiao Liao, Yihan Mu.

**Project administration:** Wei Deng.

**Resources:** Wei Deng, Hongyue Ma.

**Software:** Xiao Liao.

**Supervision:** Wei Deng, Hongyue Ma.

**Validation:** Xiao Liao, Wei Deng, Hongyue Ma.

**Visualization:** Xiao Liao.

**Writing – original draft:** Xiao Liao.

**Writing – review & editing:** Xiao Liao, Wei Deng, Hongyue Ma, Yihan Mu.

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
