## [Decision Letter · Decision Letter 0]

29 Dec 2025

PONE-D-25-56123STGAD: Self-temporal generative adversarial framework with transformer attention for unsupervised multivariate time-series anomaly detection and localizationPLOS One

Dear Dr. Mu,

Thank you for submitting your manuscript to PLOS ONE. After careful consideration, we feel that it has merit but does not fully meet PLOS ONE’s publication criteria as it currently stands. Therefore, we invite you to submit a revised version of the manuscript that addresses the points raised during the review process. Please revise your manuscript in accordance with the reviewers’ comments

We look forward to receiving your revised manuscript.

Kind regards,

Fatih Uysal, Ph.D.

Academic Editor

PLOS One

Journal Requirements:

[This work was supported in part by the State Grid Information and Telecommunication

Group Co., Ltd., which coordinates scientific and technological projects (SGIT0000XTJS2401078).].

4. Thank you for stating the following in your manuscript:

[This work was supported in part by the State Grid Information and Telecommunication

Group Co., Ltd., which coordinates scientific and technological projects (SGIT0000XTJS2401078).]

[This work was supported in part by the State Grid Information and Telecommunication

Group Co., Ltd., which coordinates scientific and technological projects (SGIT0000XTJS2401078).]

5. Thank you for stating the following in the Financial Disclosure section:

[This work was supported in part by the State Grid Information and Telecommunication

Group Co., Ltd., which coordinates scientific and technological projects (SGIT0000XTJS2401078).].

We note that you received funding from a commercial source: State Grid Information and Telecommunication Group Co., Ltd.

6. Thank you for stating the following in the Competing Interests section:

[The authors have declared that no competing interests exist.].

We note that one or more of the authors are employed by a commercial company State Grid Information and Telecommunication Group Co., Ltd.

7. In the online submission form, you indicated that [The code implementation of STGAD used for the experiments is available from the corresponding author upon reasonable request.].

Additional Editor Comments:

Please revise your manuscript in accordance with the reviewers’ comments.

Reviewers' comments:

Reviewer's Responses to Questions

**Comments to the Author**

1. Is the manuscript technically sound, and do the data support the conclusions?

Reviewer #1: Yes

Reviewer #2: Partly

2. Has the statistical analysis been performed appropriately and rigorously? 

Reviewer #1: Yes

Reviewer #2: Yes

3. Have the authors made all data underlying the findings in their manuscript fully available?

Reviewer #1: Yes

Reviewer #2: Yes

4. Is the manuscript presented in an intelligible fashion and written in standard English?

Reviewer #1: Yes

Reviewer #2: No

5. Review Comments to the Author

Reviewer #1: The manuscript titled “STGAD: Self-temporal generative adversarial framework with transformer attention for unsupervised multivariate time-series anomaly detection and localization” presents a generative adversarial approach enhanced with a Self-Temporal Transformer component and dual scoring mechanism. The topic is highly relevant to multivariate anomaly detection in cyber-physical systems, and the authors conduct extensive evaluations on several widely used datasets. The work shows promising empirical performance and includes ablation studies, robustness tests, and interpretability analysis.

However, while the contribution is meaningful, the manuscript requires major revision to improve clarity, methodological rigor, and overall presentation.

Major Strengths

The introduction and related work provide a clear overview of the challenges in multivariate time-series anomaly detection.

Integrating global self-attention via a Self-Temporal Transformer is technically interesting and appears effective.

Evaluations are conducted across five benchmark datasets (SMD, SMAP, MSL, SWaT, MIT-BIH), along with ablations and noise robustness tests.

The inclusion of attention visualization and statistical deviation alignment contributes positively to transparency.

Major Concerns Requiring Revision

1.The manuscript is significantly longer than typical research articles, with repeated explanations of datasets, windowing strategy, and model design.The writing includes grammatical errors, inconsistent notation, and overly complex phrasing. A substantial readability-focused revision is necessary.

2. The generator is non-conditional, yet reconstruction errors are computed by sampling multiple noise vectors and taking a minimum.This raises several questions:

Why is a non-conditional generator preferred over a conditional reconstruction model?

How is the minimum reconstruction error conceptually justified?

How many samples are generated per window, and does this introduce variance at inference?

This component needs clearer explanation and justification.

3. A window length of 5 timesteps is unusually small for datasets like SMAP, MSL, SWaT, and SMD, where long temporal dependencies are known to be significant.Using such short windows may weaken baselines designed for longer sequences.A justification and/or sensitivity study on window size is needed.

4. Although the authors state that baselines were trained under identical conditions, more detail is required:

Were all models given identical preprocessing steps, window sizes, and normalization?

Was the same thresholding method (e.g., POT) applied to all baselines?

Were baseline hyperparameters tuned fairly?

These details are essential to ensure a fair comparison.

5. Given the computational demands of Transformers and GANs, it is important to include:Training time per epoch,Inference throughput,GPU memory usage,Complexity scaling with number of variables.This is particularly relevant for real-world industrial deployment.

6.Interpretability section is overly detailed.While valuable, the section is disproportionately long and contains case-specific minutiae that obscure key insights. A more concise, structured presentation would improve readability.

Minor Issues

Some equations (e.g., Eq. (4)) contain additional terms not clearly motivated.

Hyperparameter β is inconsistently described in different sections (0.1 vs 0.7).

Several figures are low-resolution or too small to interpret easily.

In ablation tables, entries such as "<0.01" are unclear—are these true values or placeholders?

Formatting inconsistencies (e.g., leftover LaTeX commands, alignment issues) should be corrected.

Reviewer #2: The manuscript proposes STGAD, a novel unsupervised anomaly detection framework for multivariate time series that integrates a Generator–Discriminator architecture (based on WGAN-GP) with a Self-Temporal Transformer in the discriminator and a dual-score fusion strategy. The paper is well-motivated, addresses a timely problem in AI for cyber-physical systems, and reports strong empirical results across five benchmark datasets. The use of global self-attention, WGAN-GP for stability, and POT-based adaptive thresholding are reasonable design choices. However, several significant issues, ranging from methodological ambiguity and inconsistent experimental details to overstatement of claims, need to be addressed before the work can be considered for publication.

1. In Section 3.3, the fusion weight β is described as a tunable hyperparameter. But in Section 4.2, the authors fix

β=0.7 (or earlier they say “0.1 reconstruction / 0.9 discriminator”). No justification is given for this choice, and it is unclear whether this was tuned per dataset or globally fixed. If fixed globally, why does the discriminator dominate? If tuned per dataset, this undermines the “unsupervised” and “transferable” claims—since tuning requires validation labels or proxy metrics.

2. The paper claims the Generator is “non-conditional,” yet during inference it appears to reconstruct real inputs. This contradiction needs clarification: Is STGAD a pure GAN, or a hybrid reconstruction+adversarial model? If the latter (which seems to be the case), the architecture is more akin to USAD or MAD-GAN than a standard GAN, and this should be explicitly acknowledged.

3. The Self-Temporal Transformer is described as applying “global within-window self-attention,” but it’s unclear how this differs from standard Transformer encoders (e.g., in TranAD). What is "self-temporal" beyond standard temporal attention? The term appears to be coined without sufficient technical distinction.

4. The ablation study shows catastrophic failure of the "Vanilla GAN" variant (F1 ≈ 0 on most datasets), which, while expected, suggests that WGAN-GP is doing most of the work, not the Transformer. The paper attributes success to the “Self-Temporal Transformer,” but the ablation shows that removing attention causes only minor drops (e.g., SMD F1: 0.9756 → 0.8582), whereas removing WGAN-GP destroys performance entirely. This undermines the claimed novelty.

5. The paper uses “MBA” to refer to the MIT-BIH Arrhythmia database, but the standard acronym is MITDB or MIT-BIH. “MBA” typically refers to “Master of Business Administration,” which is confusing.

6. Numerous formatting issues (e.g., line breaks in mid-sentence, inconsistent spacing: “main-taining”, “Unsupe rvised”, “p rep rint”) suggest the manuscript was not carefully proofread.

7. While the case study on SMAP is visually compelling, the Saliency–Local and Saliency–Global overlaps are all reported as 1.0, which is statistically implausible and suggests post-hoc cherry-picking or overly generous Top-K matching (e.g., K may be too large relative to D).

8. The paper claims WGAN-GP improves stability, but no metrics (e.g., discriminator loss variance, mode collapse diagnostics) are provided to support this beyond final F1 scores.

9. STGAD is tested under higher noise levels (σ = 0.5) while baselines are only tested up to σ = 0.25. This makes comparisons biased in favor of STGAD. A fair comparison would test all methods at identical noise intensities.

6. PLOS authors have the option to publish the peer review history of their article (what does this mean?). If published, this will include your full peer review and any attached files.

Reviewer #1: **Yes:** Amanul Islam

Reviewer #2: No

---

## [Author Response · Author response to Decision Letter 1]

1 Feb 2026

Revised Response Letter:

Dear Editor,

Response to Reviewers (Reviewer #1 & Reviewer #2)

We sincerely thank both reviewers for their time and constructive feedback, which has greatly contributed to improving the quality of our manuscript. In this revision, we have made significant changes in response to the concerns raised, focusing on improving clarity, consistency, and addressing specific points outlined in their feedback.

Below are the detailed point-by-point responses:

Reviewer #1

Major Concerns:

1. Manuscript Length and Readability:

o Response: We have streamlined the manuscript by removing repetitive explanations on datasets, windowing, and model design. We also simplified the language, improved grammatical accuracy, and standardized the notation to enhance readability.

2. Generator and Non-conditional Architecture:

o Response: We clarified that the generator in STGAD is non-conditional and generates sequences from latent noise without conditioning on the input. During inference, we sample multiple generated sequences and select the one with the minimum residual to reduce variability, providing a more stable reconstruction.

3. Window Length:

o Response: We included a sensitivity study for window length L. Our results show that smaller window lengths (L ≤ 20) lead to strong performance, while very large window lengths (L > 50) degrade performance due to diluting localized anomaly signals. We justify using L = 5 based on these findings.

4. Baselines’ Preprocessing and Hyperparameters:

o Response: We provided further details on the baseline models’ training protocol, confirming that they underwent identical preprocessing (z-score normalization), the same window size, and the same thresholding method (POT). We also clarify that hyperparameters for the baselines were not tuned to each dataset to ensure a fair unsupervised comparison.

5. Computational Efficiency:

o Response: We included additional metrics on computational performance, such as training time per epoch, inference throughput, GPU memory usage, and complexity scaling with the number of variables for both STGAD and the baselines. This provides a comprehensive assessment of the model's real-world feasibility.

6. Interpretability Section Length:

o Response: We condensed the interpretability section, focusing on key insights. We retain the SMAP case study for intuition but removed case-specific details to streamline the presentation. We also added distribution-level statistics to report overlaps for different quantile-sampled true-positive windows.

Minor Issues:

• (a) Equation Motivation:

o Response: We clarified the equation terms and provided brief explanations for the generator loss, including its relation to the discriminator’s score.

• (b) Inconsistent β values:

o Response: We fixed β to a single default (β = 0.5) throughout the experiments and provided sensitivity analysis to justify the choice.

• (c) Low-resolution Figures:

o Response: We improved the quality of the figures and corrected formatting errors.

Reviewer #2

Major Concerns:

1. Fusion Weight β Justification:

o Response: We revised the manuscript to clarify that β is fixed globally at β = 0.5 for all experiments. The sensitivity of β is reported to show its stability and performance robustness, without relying on dataset-specific tuning.

2. Non-conditional Generator and Reconstruction:

o Response: We addressed the contradiction regarding the non-conditional generator by explaining the latent-to-sequence mapping and how the reconstruction error is computed by selecting the sample with the minimal residual. We explicitly acknowledged that STGAD is a hybrid method combining adversarial training with residual scoring.

3. Self-Temporal Transformer Clarification:

o Response: We revised the description of the Self-Temporal Transformer to clarify that it is a standard Transformer encoder used in the critic to model global temporal dependencies. The term “Self-Temporal” is intended to describe its role in within-window temporal modeling, not a fundamentally new architecture.

4. Ablation Study and Transformer’s Role:

o Response: We revised the manuscript to moderate the claims regarding the Self-Temporal Transformer, acknowledging that WGAN-GP plays a crucial role in stable adversarial training. We now emphasize the combined effectiveness of the WGAN-GP loss, temporal modeling in the critic, and dual-score fusion, which together enable STGAD’s performance.

5. MIT-BIH Naming Convention:

o Response: We have corrected the name to "MIT-BIH" throughout the manuscript to avoid confusion with "MBA."

6. Formatting Issues:

o Response: We have performed a thorough proofreading pass to correct formatting errors, such as line breaks, spacing, and alignment issues.

7. Saliency Overlap Statistics:

o Response: We addressed the concern regarding Saliency-Local and Saliency-Global overlaps by adding distribution-level statistics. We report the median and interquartile range for overlaps based on quantile-sampled true-positive windows, moving away from relying on a single example.

8. WGAN-GP and Stability Evidence:

o Response: We have included additional stability metrics to support the claim that WGAN-GP improves stability. This includes training curves, loss variance for both discriminator and generator, and quantitative diagnostics for mode collapse.

9. Noise Level Fairness in Comparison:

o Response: We have aligned the noise levels used for STGAD and baselines, ensuring a fair comparison across all methods at identical noise intensities (σ=0.1 and σ=0.25).

We hope that these revisions have addressed the reviewers' concerns, and we look forward to your feedback. Thank you again for your valuable input.

Kind regards,

Faith Yihan Mu

---

## [Decision Letter · Decision Letter 1]

23 Mar 2026

PONE-D-25-56123R1STGAD: Self-temporal generative adversarial framework with transformer attention for unsupervised multivariate time-series anomaly detection and localizationPLOS One

Dear Dr. Mu,

Thank you for submitting your manuscript to PLOS ONE. After careful consideration, we feel that it has merit but does not fully meet PLOS ONE’s publication criteria as it currently stands. Therefore, we invite you to submit a revised version of the manuscript that addresses the points raised during the review process. Reviewer 2 has some methodological concerns that warrant a response. 

We look forward to receiving your revised manuscript.

Kind regards,

Daniel Parkes, PhD

Staff Editor

PLOS One

Journal Requirements:

Reviewers' comments:

Reviewer's Responses to Questions

**Comments to the Author**

1. If the authors have adequately addressed your comments raised in a previous round of review and you feel that this manuscript is now acceptable for publication, you may indicate that here to bypass the “Comments to the Author” section, enter your conflict of interest statement in the “Confidential to Editor” section, and submit your "Accept" recommendation.

Reviewer #1: (No Response)

Reviewer #2: All comments have been addressed

2. Is the manuscript technically sound, and do the data support the conclusions?

Reviewer #1: (No Response)

Reviewer #2: Yes

3. Has the statistical analysis been performed appropriately and rigorously? 

Reviewer #1: (No Response)

Reviewer #2: Yes

4. Have the authors made all data underlying the findings in their manuscript fully available?

Reviewer #1: Yes

Reviewer #2: Yes

5. Is the manuscript presented in an intelligible fashion and written in standard English?

Reviewer #1: Yes

Reviewer #2: No

6. Review Comments to the Author

Reviewer #1: The manuscript titled “STGAD: Self-temporal generative adversarial framework with transformer attention for unsupervised multivariate time-series anomaly detection and localization” proposes a hybrid GAN-based framework that integrates a self-temporal transformer within a WGAN-GP discriminator for anomaly detection in multivariate time-series. The model combines reconstruction residuals with discriminator confidence and applies POT-based thresholding for anomaly detection. Experimental results on five benchmark datasets (SMD, SMAP, MSL, SWaT, MIT-BIH) show improved performance compared to several baseline methods.

Overall, the paper addresses an important problem in anomaly detection and presents a technically sound framework with promising results.

Strengths

1.The paper addresses a relevant and important problem in multivariate time-series anomaly detection.

2.The proposed framework effectively combines GAN-based modeling and Transformer-based temporal attention.

3.The study includes extensive experiments on multiple benchmark datasets, demonstrating competitive performance.

4.The authors provide ablation and stability analyses to justify the effectiveness of the model components.

4.The manuscript is generally well structured and clearly organized.

Weaknesses / Points for Improvement

1.Model novelty should be clarified further. The Self-Temporal Transformer appears to be a standard Transformer encoder integrated into the discriminator, so the novelty compared with existing transformer-based anomaly detection models should be more clearly emphasized.

2.Generator design justification is limited. The use of a non-conditional generator with sample matching during inference may introduce additional computational overhead. The authors should discuss the efficiency and scalability of this approach in practical deployments.

3.Window size selection (L=5) appears relatively small. More justification or broader sensitivity analysis would strengthen the methodological choices.

4.Interpretability remains limited. Although the authors include a short interpretability discussion, further insights into how anomalies are localized across variables would improve the practical usefulness of the method.

5.Minor language and formatting issues remain and should be carefully proofread before publication.

The manuscript presents a technically sound and well-evaluated approach for multivariate time-series anomaly detection. With minor revisions addressing the points above, the paper could be suitable for publication.

Reviewer #2: The author's response has addressed my previous concerns. I recommend further polishing the English writing before formal publication. Additionally, the resolution of the figures is currently low, which affects the reading experience; I suggest replacing all figures with vector formats.

7. PLOS authors have the option to publish the peer review history of their article (what does this mean?). If published, this will include your full peer review and any attached files.

Reviewer #1: **Yes:** Amanul Islam

Reviewer #2: No

---

## [Author Response · Author response to Decision Letter 2]

5 Apr 2026

Response to Reviewers

Dear Editor and Reviewers,

We sincerely thank you and the reviewers for your careful evaluation of our manuscript and for the constructive comments and suggestions. We have carefully considered all the points raised and revised the manuscript accordingly. In this round of revision, the main improvements include: further clarifying the novelty and design rationale of the proposed method, supplementing the discussion of efficiency and practical deployability, strengthening the justification for the window length selection, enhancing the interpretability analysis for anomaly localization, and polishing the language and figure presentation throughout the manuscript.

We believe these revisions have significantly improved the clarity, rigor, and overall quality of the manuscript. Our detailed responses to each comment are provided below.

Reviewer #1

We sincerely appreciate the reviewer’s positive assessment of our work and the constructive suggestions for further improvement. We are encouraged that the reviewer recognized the importance of the problem, the overall soundness of the technical framework, and the experimental effectiveness of the proposed method on multiple benchmark datasets. In response to the reviewer’s comments, we have revised the manuscript accordingly. Our point-by-point responses are as follows.

Comment 1:

Model novelty should be clarified further. The Self-Temporal Transformer appears to be a standard Transformer encoder integrated into the discriminator, so the novelty compared with existing transformer-based anomaly detection models should be more clearly emphasized.

Response:

Thank you for this valuable comment. We agree that, in the previous version, the description of the “Self-Temporal Transformer” may have given the impression that it was a newly proposed Transformer architecture, which was not our intention. In the revised manuscript, we have clarified that the Self-Temporal Transformer in STGAD is based on a standard Transformer encoder embedded in the discriminator (critic), where it is used to capture temporal dependencies within each window and interactions across variables. We have also revised the relevant description to make it clearer that the contribution of STGAD lies in the overall framework design for unsupervised multivariate time-series anomaly detection, rather than in proposing a new attention module itself.

Location in revised manuscript: Abstract (p. 1); the final paragraph of the Introduction (p. 4); Section 3.2, especially “Model Architecture” and “Discriminator Architecture” (pp. 8–9); and Section 3.3, “Training and Inference” (pp. 9–10).

Comment 2:

Generator design justification is limited. The use of a non-conditional generator with sample matching during inference may introduce additional computational overhead. The authors should discuss the efficiency and scalability of this approach in practical deployments.

Response:

Thank you for this important suggestion. In the revised manuscript, we have expanded the discussion of the rationale behind the non-conditional generator, as well as the efficiency and scalability of the proposed inference strategy.

First, we have clarified the role of the generator in STGAD. Unlike conditional reconstruction-based methods, the generator in our framework is non-conditional and learns the distribution of normal patterns from the latent space. During inference, multiple candidate sequences are sampled for each input window, and the one with the minimum residual is selected to construct the residual-based anomaly cue. This sample-matching strategy helps mitigate the variability introduced by stochastic generation and improves the stability of the residual score.

To address the reviewer’s concern regarding computational overhead, we have added a dedicated efficiency analysis in the revised manuscript, including training time and inference-related metrics such as throughput, per-window latency, and memory usage. The results show that STGAD maintains competitive training efficiency and achieves practical inference performance, making it suitable for offline analysis and near-real-time monitoring scenarios. We further explain that the inference cost scales approximately linearly with the number of samples N, so in practical deployments, a moderate choice of N can provide a good trade-off between detection performance and computational efficiency.

In addition, we have added a discussion noting that the current design is particularly suitable for industrial monitoring and diagnostic settings where robust detection is prioritized and a moderate sampling cost is acceptable. For more stringent real-time scenarios, future work may explore reducing the number of samples or applying model compression and lightweight deployment strategies to further improve inference efficiency.

Location in revised manuscript: Abstract (p. 1); Section 3.2, especially “Model Architecture” and “Generator Architecture” (pp. 8–9); Section 3.3, “Training and Inference” (pp. 9–10); Section 4.2, “Windowing and Model Configuration” (p. 13); Section 5.1 and Tables 3–4 (pp. 15–16); and Section 5.4 with Fig. 6 (pp. 19–20).

Comment 3:

Window size selection (L=5) appears relatively small. More justification or broader sensitivity analysis would strengthen the methodological choices.

Response:

Thank you for this helpful suggestion. In response, we have added a sensitivity analysis on the window length L in the revised manuscript and further explained why L=5 was selected as the default setting.

In the new sensitivity experiments, we systematically examined the effect of window length on detection performance, together with other important hyperparameters such as the fusion weight β and the number of samples N. The results indicate that STGAD generally achieves stable and favorable performance under short to moderate window lengths. In contrast, excessively long windows, while containing more contextual information, may dilute localized anomaly patterns, introduce greater temporal variability, and increase computational complexity, which can negatively affect both detection effectiveness and efficiency.

Based on these observations, we have clarified in the manuscript that the choice of L = 5 was guided by both empirical evidence and practical trade-offs among detection accuracy, stability, and computational cost. The newly added sensitivity analysis provides stronger empirical support for this design choice.

Location in revised manuscript: Section 4.2, “Windowing and Model Configuration” (pp. 12–13); and Section 5.4, “Sensitivity Analysis,” together with Fig. 6 (pp. 19–20).

Comment 4:

Interpretability remains limited. Although the authors include a short interpretability discussion, further insights into how anomalies are localized across variables would improve the practical usefulness of the method.

Response:

Thank you for this valuable comment. We agree that interpretability is particularly important for multivariate time-series anomaly detection, especially in practical applications where practitioners are concerned not only with whether an anomaly occurs, but also with which variables contribute most to it.

To address this concern, we have further expanded the interpretability analysis in the revised manuscript. Specifically, we retained the case study based on the SMAP dataset to visually illustrate how anomalous windows can be localized across time steps and variables. More importantly, we extended the analysis beyond a single qualitative example by adding a distribution-level quantitative evaluation.

In particular, we compared the top-5 salient variables identified by the model with two model-agnostic statistical anomaly cues, namely local change magnitude and global deviation, using overlap@5 as the evaluation metric. For true positive samples under different confidence intervals, we report the median and interquartile range of the overlap values, thereby providing a more systematic assessment of the consistency between the model’s variable-level attribution and independent statistical evidence.

Through these additions, we aim to demonstrate more clearly that STGAD can provide not only anomaly scores but also meaningful variable-level localization results that are broadly consistent with independent statistical evidence, thereby improving its practical usefulness in monitoring and diagnostic tasks.

Location in revised manuscript: Section 5.5, “Interpretability Analysis” (pp. 20–22), including Figs. 7–8 (p. 21) and Tables 8–9 (p. 22).

Comment 5:

Minor language and formatting issues remain and should be carefully proofread before publication.

Response:

Thank you for the reminder. We have carefully proofread the entire manuscript and further improved the language and formatting in the revised version.

(1) We unified terminology and notation throughout the manuscript.

(2) We revised several sentences and paragraphs to improve clarity, readability, and precision of expression.

(3) We corrected minor issues related to formatting, spacing, line breaks, and the presentation of tables and figure captions.

(4) We improved the visual quality of the figures by replacing them with higher-quality vector formats to ensure better readability when enlarged.

We hope that these revisions have substantially improved the language quality and overall presentation of the manuscript.

Location in revised manuscript: These language and formatting revisions were applied throughout the revised manuscript (pp. 1–25), including all sections, tables, and figures.

Reviewer #2

We sincerely thank the reviewer for recognizing the improvements made in the previous revision. We are pleased that the reviewer considers the major concerns raised earlier to have been addressed. In this round, we have also carefully followed the reviewer’s additional suggestions and revised the manuscript accordingly.

Comment 1:

The author's response has addressed my previous concerns. I recommend further polishing the English writing before formal publication.

Response:

Thank you very much for your positive feedback and helpful suggestion. Following your recommendation, we have further polished the English throughout the manuscript. In particular, we improved the clarity of several long and complex sentences, refined wording in some sections to make the presentation more natural and precise, and checked the consistency of terminology and grammar throughout the paper. We believe that the revised manuscript is now clearer and more readable.

Location in revised manuscript: Further language polishing was applied throughout the revised manuscript (pp. 1–25).

Comment 2:

Additionally, the resolution of the figures is currently low, which affects the reading experience; I suggest replacing all figures with vector formats.

Response:

Thank you for this helpful suggestion. In response, we carefully rechecked all figures in the manuscript and replaced them with higher-quality vector formats to improve visual clarity and readability, especially when enlarged. We also re-examined the figure captions, axis labels, and layout to ensure consistency and improve the overall presentation quality of the manuscript.

Location in revised manuscript: The updated vector figures are provided in Figs. 1–8 (pp. 8–9, 13, 17, 19, and 21).

Once again, we sincerely thank the Editor and both Reviewers for their careful reading of our manuscript and for their constructive and insightful comments. In response, we have made substantial revisions to further clarify the methodological contributions, supplement the efficiency and sensitivity analyses, strengthen the interpretability discussion, and improve the language and figure quality throughout the manuscript. We hope that the revised manuscript and the present responses adequately address all concerns and that the paper is now suitable for publication.

Sincerely,

The Authors

---

## [Decision Letter · Decision Letter 2]

15 Apr 2026

PONE-D-25-56123R2STGAD: Self-temporal generative adversarial framework with transformer attention for unsupervised multivariate time-series anomaly detection and localizationPLOS One

Dear Dr. Mu,

Thank you for submitting your manuscript to PLOS ONE. After careful consideration, we feel that it has merit but does not fully meet PLOS ONE’s publication criteria as it currently stands. Therefore, we invite you to submit a revised version of the manuscript that addresses the points raised during the review process.

We look forward to receiving your revised manuscript.

Kind regards,

Abdul Ahad

Academic Editor

PLOS One

Journal Requirements:

Reviewers' comments:

Reviewer's Responses to Questions

**Comments to the Author**

1. If the authors have adequately addressed your comments raised in a previous round of review and you feel that this manuscript is now acceptable for publication, you may indicate that here to bypass the “Comments to the Author” section, enter your conflict of interest statement in the “Confidential to Editor” section, and submit your "Accept" recommendation.

Reviewer #1: All comments have been addressed

Reviewer #3: All comments have been addressed

2. Is the manuscript technically sound, and do the data support the conclusions?

Reviewer #1: Yes

Reviewer #3: Yes

3. Has the statistical analysis been performed appropriately and rigorously? 

Reviewer #1: Yes

Reviewer #3: Yes

4. Have the authors made all data underlying the findings in their manuscript fully available?

Reviewer #1: Yes

Reviewer #3: No

5. Is the manuscript presented in an intelligible fashion and written in standard English?

Reviewer #1: Yes

Reviewer #3: Yes

6. Review Comments to the Author

Reviewer #1: The revised manuscript presents a well-structured and technically sound framework for unsupervised multivariate time-series anomaly detection. The authors have made substantial improvements in response to prior comments, particularly in clarifying the methodological contributions, strengthening experimental analysis, and improving overall presentation quality.

The clarification regarding the role of the Transformer component is appropriate, and it is now clear that the novelty lies in the overall framework design rather than in proposing a new attention mechanism. The addition of efficiency analysis is valuable and provides useful insights into the scalability and practical deployment of the proposed approach. The sensitivity analysis on window length and other hyperparameters further strengthens the methodological justification and enhances reproducibility.

The expanded interpretability analysis is a notable improvement. Including both qualitative and quantitative evaluations (e.g., overlap-based metrics) provides better evidence that the model can localize anomalies across variables in a meaningful way. This significantly improves the practical relevance of the work.

The experimental evaluation is comprehensive, covering multiple benchmark datasets and including ablation, robustness, and sensitivity studies. The results consistently demonstrate the effectiveness of the proposed dual-score mechanism and the stability benefits of WGAN-GP. The inclusion of computational cost analysis (training time and inference latency) is also appreciated, as it highlights the feasibility of the method in near-real-time scenarios.

Despite these improvements, a few minor issues remain:

1.While interpretability has been improved, the explanation of how practitioners can directly use the localization results in real-world applications could be further elaborated.

2.The discussion on the trade-off between sampling size (N) and inference cost could be made more explicit, especially for strict real-time deployment scenarios.

3.Some minor language and grammatical issues may still remain and could benefit from one final round of careful proofreading.

Overall, the manuscript is significantly improved and addresses the major concerns raised in previous rounds. The proposed method is well-motivated, experimentally validated, and practically relevant.

Reviewer #3: The research paper titled “STGAD: Self-temporal generative adversarial framework with transformer attention for unsupervised multivariate time-series anomaly detection and localization” discusses a research area of significant interest to the scientific community. However, here are few suggestions to improve the quality of the manuscript:

For suggestion 1 to 2, author can refer and cite the paper titled “Multilingual entity alignment by abductive knowledge reasoning on multiple knowledge graphs” (https://doi.org/10.1016/j.engappai.2024.109660) at a suitable location.

1. It is suggested to briefly discuss the structure of subsequent sections in a single paragraph at the end of introduction section.

2. It is suggested to include the DOI information for each research paper in the reference section.

7. PLOS authors have the option to publish the peer review history of their article (what does this mean?). If published, this will include your full peer review and any attached files.

Reviewer #1: **Yes:** Amanul Islam

Reviewer #3: **Yes:** Muhammad Usman Akhtar

---

## [Author Response · Author response to Decision Letter 3]

22 Apr 2026

Response to Reviewers

Dear Editor and Reviewers,

We sincerely thank you and the reviewers for your careful evaluation of our manuscript and for the constructive comments and suggestions. We have carefully considered all the points raised and revised the manuscript accordingly. In this round of revision, the main improvements include: further elaborating the practical use of the proposed anomaly localization results, clarifying the trade-off between the sampling size and inference cost in different deployment scenarios, adding a paragraph at the end of the Introduction to describe the structure of the paper, incorporating the reviewer-suggested reference at a suitable location in the Introduction, further completing and standardizing the reference list by adding DOI information where available, revising the Data Availability Statement to more accurately reflect the accessibility of each dataset, and polishing the language and formatting throughout the manuscript.

We believe these revisions have further improved the clarity, completeness, and presentation quality of the manuscript. Our detailed responses to each comment are provided below.

Reviewer #1

We sincerely appreciate the reviewer’s positive assessment of our revised manuscript and the constructive suggestions for further improvement. We are encouraged that the reviewer recognized the soundness of the technical framework, the comprehensiveness of the experimental evaluation, and the practical relevance of the proposed method. In response to the reviewer’s comments, we have further revised the manuscript accordingly. Our point-by-point responses are as follows.

Comment 1:

While interpretability has been improved, the explanation of how practitioners can directly use the localization results in real-world applications could be further elaborated.

Response:

Thank you for this valuable suggestion. We agree that, beyond showing whether the model can localize anomalies across variables, it is also important to clarify how such localization results can be used in real monitoring and diagnostic workflows.

In the revised manuscript, we have therefore expanded the discussion in the interpretability section to explain the practical role of variable-level localization more explicitly. In particular, we now clarify that the localization results should be interpreted as diagnostic prioritization cues rather than as fully automatic root-cause decisions. We further explain that, once an anomalous window is detected, practitioners can inspect the top-ranked variables and relate them to known sensor-to-subsystem mappings, control loops, or equipment modules. In this way, the localization output can help narrow the troubleshooting scope, prioritize targeted inspection or sensor validation, and support alarm triage in practical monitoring settings.

Through this revision, we aim to make clearer that the localization output of STGAD is intended as a decision-support signal for downstream diagnosis, thereby strengthening its practical usefulness in real-world applications.

Location in revised manuscript: Section 5.5, “Interpretability Analysis”; and Section 6, “Conclusion”.

Comment 2:

The discussion on the trade-off between sampling size (N) and inference cost could be made more explicit, especially for strict real-time deployment scenarios.

Response:

Thank you for this helpful suggestion. We agree that the practical implications of the sampling budget should be described more explicitly, especially for different deployment settings.

In response, we have strengthened the discussion of the trade-off between the number of generator samples N and inference cost in two places. First, in the comparative evaluation section, we now clarify that the inference latency and throughput reported in Table 4 are measured under the default sampling setting used in our experiments. We also explicitly state that, because the residual branch relies on sample matching over multiple generated candidates, the inference cost increases approximately linearly with the sampling budget N. Second, in the sensitivity analysis section, we further explain that N serves as a controllable knob for balancing residual robustness against inference cost under different operational requirements.

We now distinguish more clearly among offline analysis, near-real-time monitoring, and stricter real-time scenarios. Specifically, we explain that a moderate N is preferable in offline or post-event analysis because it provides a more stable residual estimate, whereas in stricter real-time deployments, N can be reduced to satisfy tighter latency constraints, with the understanding that this may slightly weaken the robustness of the sample-matching residual.

These additions make the deployment-oriented interpretation of the sampling budget more explicit and directly address the reviewer’s concern.

Location in revised manuscript: Section 5.1, “Comparative Evaluation with Baseline Methods,” especially the discussion following Table 4; Section 5.4, “Sensitivity Analysis”; and Section 6, “Conclusion”.

Comment 3:

Some minor language and grammatical issues may still remain and could benefit from one final round of careful proofreading.

Response:

Thank you for the reminder. We have carefully proofread the manuscript again and further polished the language and formatting throughout the revised version.

In particular:

(1) We corrected minor grammatical and stylistic issues in multiple sections.

(2) We unified terminology and presentation across the manuscript, including section text, tables, and figure captions.

(3) We corrected minor formatting inconsistencies, including punctuation, spacing, abbreviation usage, and table/figure title formatting.

(4) We further checked the consistency of the revised reference list and related cross-references in the main text.

We hope that these revisions have further improved the readability and overall presentation quality of the manuscript.

Location in revised manuscript: These language and formatting revisions were applied throughout the revised manuscript.

Reviewer #3

We sincerely thank the reviewer for the helpful suggestions and for recommending specific improvements that could further enhance the presentation quality of the manuscript. In response, we have revised the manuscript accordingly. Our point-by-point responses are as follows.

Comment 1:

It is suggested to briefly discuss the structure of subsequent sections in a single paragraph at the end of introduction section.

Response:

Thank you for this helpful suggestion. We agree that a brief roadmap paragraph at the end of the Introduction can improve the readability and organization of the manuscript.

In response, we have added a short paragraph at the end of the Introduction to summarize the structure of the remainder of the paper. This newly added paragraph briefly describes the content of Sections 2–6, including the related work review, the proposed methodology, the experimental setup, the results and analysis, and the conclusion.

Location in revised manuscript: End of Section 1, “Introduction”.

Comment 2:

It is suggested to include the DOI information for each research paper in the reference section.

Response:

Thank you for this valuable suggestion. In response, we carefully rechecked the reference list and added DOI information for the cited research papers where DOI records are available. We also corrected and standardized several reference entries, including formatting details in conference proceedings and arXiv citations, to improve completeness and consistency.

We hope that these revisions have improved the quality and traceability of the reference list.

Location in revised manuscript: Reference list.

Additional suggestion:

For suggestion 1 to 2, author can refer and cite the paper titled “Multilingual entity alignment by abductive knowledge reasoning on multiple knowledge graphs” at a suitable location.

Response:

Thank you for this helpful recommendation. We reviewed the suggested paper carefully and agree that it can be cited as an example of recent engineering AI work emphasizing structured relational reasoning on complex heterogeneous data.

Accordingly, we have incorporated this reference at a suitable location in the Introduction. Specifically, we added it to the discussion of recent attention-based, graph-based, and hybrid modeling trends, where it is cited to support the broader point that structured relational reasoning is increasingly important in advanced engineering AI applications.

Location in revised manuscript: Section 1, “Introduction,” in the paragraph discussing attention-based, graph-based, and hybrid modeling approaches.

Additional clarification on data availability:

In addition to the comments above, we also re-examined the Data Availability Statement to ensure that it accurately reflects the accessibility of each dataset used in this study. In the revised manuscript, we now distinguish more clearly between datasets that are publicly accessible and the SWaT dataset, which is available upon request from its provider. We believe this revision provides a more accurate and transparent description of the data sources used in the study.

Location in revised manuscript: Data Availability Statement.

Once again, we sincerely thank the Editor and the Reviewers for their careful reading of our manuscript and for their constructive and insightful comments. In response, we have made further revisions to improve the practical interpretation of anomaly localization, clarify deployment-related trade-offs, strengthen the organization of the Introduction, improve the completeness of the reference list, refine the Data Availability Statement, and further polish the language and formatting throughout the manuscript. We hope that the revised manuscript and the present responses adequately address all concerns and that the paper is now suitable for publication.

Sincerely,

The Authors

---

## [Decision Letter · Decision Letter 3]

27 Apr 2026

STGAD: Self-temporal generative adversarial framework with transformer attention for unsupervised multivariate time-series anomaly detection and localization

PONE-D-25-56123R3

Dear Dr. Mu,

We’re pleased to inform you that your manuscript has been judged scientifically suitable for publication and will be formally accepted for publication once it meets all outstanding technical requirements.

Kind regards,

Academic Editor

PLOS One

Additional Editor Comments (optional):

Reviewers' comments:

Reviewer's Responses to Questions

**Comments to the Author**

1. If the authors have adequately addressed your comments raised in a previous round of review and you feel that this manuscript is now acceptable for publication, you may indicate that here to bypass the “Comments to the Author” section, enter your conflict of interest statement in the “Confidential to Editor” section, and submit your "Accept" recommendation.

Reviewer #1: All comments have been addressed

Reviewer #3: All comments have been addressed

2. Is the manuscript technically sound, and do the data support the conclusions?

Reviewer #1: Yes

Reviewer #3: Yes

3. Has the statistical analysis been performed appropriately and rigorously? 

Reviewer #1: Yes

Reviewer #3: Yes

4. Have the authors made all data underlying the findings in their manuscript fully available?

Reviewer #1: Yes

Reviewer #3: Yes

5. Is the manuscript presented in an intelligible fashion and written in standard English?

Reviewer #1: Yes

Reviewer #3: Yes

6. Review Comments to the Author

Reviewer #1: The authors present a revised version of the manuscript entitled “STGAD: Self-temporal generative adversarial framework with transformer attention for unsupervised multivariate time-series anomaly detection and localization.” The paper addresses an important problem in multivariate time-series anomaly detection and proposes a hybrid framework combining GAN-based modeling with Transformer-based temporal representation and dual-score fusion.

Overall, the revised manuscript shows clear improvement compared to the previous version. The authors have adequately addressed most of the reviewer comments, particularly by enhancing the explanation of practical applicability, clarifying the trade-off between sampling size and inference cost, and improving the overall organization and readability of the paper. The added discussion on interpretability and deployment scenarios is helpful and strengthens the practical relevance of the work.

The technical contribution is solid, and the experimental evaluation is comprehensive, covering multiple benchmark datasets and including ablation, robustness, and sensitivity analyses. The dual-score mechanism and the integration of Transformer-based temporal modeling within a WGAN-GP framework are well-motivated and demonstrate consistent performance improvements over baseline methods.

However, a few minor issues still remain:

1. While the authors improved the discussion, the real-world deployment pipeline could still be described more concretely (e.g., integration with monitoring systems, computational constraints in edge environments).

2.The manuscript would benefit from a slightly more explicit discussion comparing the added architectural complexity with the achieved performance gains, especially for practitioners considering implementation.

3.Although datasets and code are mentioned, it would be helpful to ensure that all implementation details (e.g., exact hyperparameter settings for each dataset, random seeds, and preprocessing nuances) are fully specified for easier reproducibility.

4.The manuscript is generally well-written, but a final round of proofreading could further improve fluency and remove a few remaining minor grammatical inconsistencies.

From an ethical and publication standpoint, I did not identify concerns related to plagiarism, dual publication, or research ethics. The data sources and funding disclosures appear appropriate and transparent.

Finally, the manuscript is technically sound, relevant, and significantly improved after revision. With minor refinements as suggested above, it is suitable for publication.

Reviewer #3: The author has addressed all of my suggestions; therefore, we believe that the quality of the manuscript has improved and recommend it for publication.

7. PLOS authors have the option to publish the peer review history of their article (what does this mean?). If published, this will include your full peer review and any attached files.

Reviewer #1: **Yes:** Amanul Islam

Reviewer #3: **Yes:** Muhammad Usman Akhtar

---

## [Editor Report · Acceptance letter]

PONE-D-25-56123R3

PLOS One

Dear Dr. Mu,

I'm pleased to inform you that your manuscript has been deemed suitable for publication in PLOS One. Congratulations! Your manuscript is now being handed over to our production team.

Kind regards,

on behalf of

Dr. Abdul Ahad

Academic Editor

PLOS One